# Phase-engineered synthesis of atomically thin te single crystals with high on-state currents

Jun Zhou[1,4], Guitao Zhang[1,4], Wenhui Wang[1,4], Qian Chen[1], Weiwei Zhao[1], Hongwei Liu[2], Bei Zhao [1] ✉, Zhenhua Ni [1,3] ✉ & Junpeng Lu [1,3] ✉

Multiple structural phases of tellurium (Te) have opened up various opportunities for the development of two-dimensional (2D) electronics and optoelectronics. However, the phase-engineered synthesis of 2D Te at the atomic level remains a substantial challenge. Herein, we design an atomic cluster density and interface-guided multiple control strategy for phase- and thickness-controlled synthesis of $\alpha$-Te nanosheets and $\beta$-Te nanoribbons (from monolayer to tens of μm) on $WS_2$ substrates. As the thickness decreases, the $\alpha$-Te nanosheets exhibit a transition from metallic to n-type semiconducting properties. On the other hand, the $\beta$-Te nanoribbons remain p-type semiconductors with an ON-state current density ($I_{ON}$) up to ~ 1527 μA μm$^{-1}$ and a mobility as high as ~ 690.7 cm$^2$ V$^{-1}$ s$^{-1}$ at room temperature. Both Te phases exhibit good air stability after several months. Furthermore, short-channel (down to 46 nm) $\beta$-Te nanoribbon transistors exhibit remarkable electrical properties ($I_{ON}$ = ~ 1270 μA μm$^{-1}$ and ON-state resistance down to 0.63 kΩ μm) at $V_{ds}$ = 1 V.

Tellurium (Te) has recently attracted much attention in optoelectronic devices due to its unique helical chain structure[1], unusual anisotropic crystal structure[1], tunable bandgap[2] and unusually low thermal conductivity[3]. Two-dimensional (2D) Te may have different crystal structures with a variety of bonding configurations[4]. Among them, the $\alpha$-Te exhibits a mixture of triplet and sixfold coordination, while $\beta$-Te exhibits a mixture of triplet and quadruple coordination, which have very different physical and electronic properties[5]. The $\alpha$-Te configuration corresponds structurally to the 1 T configuration commonly adopted for 2D materials known as transition metal dichalcogenides (TMDs), which possess inversion symmetry and semimetallic electronic structures with complex topological states[6]. Moreover, $\alpha$-Te exhibits low exciton binding energy (~ 0.18 eV), extraordinary mobility, high optical absorption, and strong infrared oscillation intensity, which holds broad prospects for applications in

sensitive photodetectors[4,7]. $\beta$-Te is a p-type semiconductor with a bandgap of ~0.3 eV, and due to spin-orbit coupling[8], it possesses a small effective mass and a high hole mobility of up to several thousand cm$^2$ V$^{-1}$ s$^{-1}$. Meantime, $\beta$-Te possesses rich and intriguing characteristics, such as photoconductivity, thermoelectricity, and piezoelectricity, which demonstrates significant potential applications in various fields, including photodetectors, field-effect transistors, piezoelectric devices, modulators, and energy harvesting devices[1]. The dynamic control of $\alpha$-Te and $\beta$-Te phase engineering can reveal the competition, coexistence, and cooperation among different crystal structures as well as the interactions between different physical properties[9]. Therefore, phase engineering between semimetallic $\alpha$-Te and semiconducting $\beta$-Te polymorphs holds significant importance for widespread device applications, such as phase-engineering memory devices, high-performance transistors,

[1]School of Physics and Key Laboratory of Quantum Materials and Devices of Ministry of Education, Southeast University, Nanjing 211189, China. [2]Jiangsu Key Lab on Opto-Electronic Technology, School of Physics and Technology, Nanjing Normal University, 1 Wenyuan Road, Nanjing 210023, China. [3]School of Electronic Science and Engineering, Southeast University, Nanjing 210096, China. [4]These authors contributed equally: Jun Zhou, Guitao Zhang, Wenhui Wang. ✉e-mail: beizhao@seu.edu.cn; zhni@seu.edu.cn; phyljp@seu.edu.cn

reconfigurable circuits, and topological transistors, under atomically thin limits[10–12].

To date, p-type $\beta$-Te materials with excellent electrical properties have been successfully synthesized using various methods, such as hydrothermal synthesis[8,13], molecular beam epitaxy (MBE)[14,15], thermal evaporation[16,17], atomic layer deposition (ALD)[18], and chemical vapor deposition (CVD)[19,20]. Although Wu et al. synthesized $\beta$-Te nanosheets with mobility of approximately 700 cm$^2$ V$^{-1}$ s$^{-1}$ and a significant ON-state current density of 1.06 mA $\mu$m$^{-1}$ at drain source voltage ($V_{ds}$) of −1.4 V using a substrate-free solution[13], their ON/OFF ratio was only 10$^1$. In addition, Zhao et al. reported the fabrication of $\beta$-Te thin films by thermal evaporation at room temperature[17] with an ON/OFF current ratio of ~10$^4$, but an average mobility was 21.1 cm$^2$ V$^{-1}$ s$^{-1}$, an ON-state current was approximately 0.001 mA $\mu$m$^{-1}$. However, these previous reports have been limited to the synthesis of $\beta$-Te, and phase engineering between semimetallic $\alpha$-Te and semiconducting $\beta$-Te has not been reported to date, especially at the monolayer limit. Exploring phase transition competition at the atomic scale is crucial because it involves significantly enhanced electron-phonon and electron-electron interactions. While numerous strategies for phase engineering 2D materials have been developed in recent years, mainly including charge injection, plasma treatment, strain application, and intercalation of alkali metal ions via electron beams[10,21–23], which often result in thermodynamically unstable phases with low purity. Furthermore, unexpected dopants or defects introduced by high external energy in phase engineering would sacrifice the intrinsic properties of different phases. Therefore, developing controlled phase engineering strategies that can produce high-purity, stable phases of 2D materials without sacrificing their intrinsic properties is important.

van der Waals (vdW) epitaxy is a recently developed technique that allows controlled synthesis of 2D materials with specific geometries or properties[24,25]. For example, the designed growth of large 2D single crystals such as graphene, h-BN, and MoS$_2$ has been realized by manipulating the energetically preferred orientation of the nuclei[26,27], which is enabled by interlayer vdW interactions related to the surface symmetry of the substrate and the orientation of the produced domains[28–33]. In addition, vdW interface interactions are applicable for controlling the interfacial formation energy between substrate and produced cluster, providing a promising approach for precise, phase-controlled synthesis of 2D materials due to the different formation energies of different phases on the substrate.

In this work, phase engineering of atomically thin $\alpha$-Te nanosheets and $\beta$-Te nanoribbons (Supplementary Fig. 1) with high purity and crystallinity was achieved via a vdW epitaxy method. Phase control is achieved by thermodynamic competition induced by atomic cluster density and interface interactions. WS$_2$ monolayers work as a substrate to provide a platform for vdW epitaxy and induce the formation of $\alpha$-Te nanosheets with a symmetrical hexagonal phase and $\beta$-Te nanoribbons with a helical tetragonal phase. Synergistic control of the phase and thickness of $\alpha$-Te and $\beta$-Te is realized for the first time. Benefiting from the quantum confinement effect, the $\alpha$-Te nanosheets transform from a metal to an n-type semiconductor with decreasing thickness, while the $\beta$-Te nanoribbons are p-type semiconductors with hole mobilities as high as ~690.7 cm$^2$ V$^{-1}$ s$^{-1}$ and a high $I_{ON}$ of 1.27 mA $\mu$m$^{-1}$ at $V_{ds}$ = −1 V. Additionally, both of them have good air stability after 5-6 months.

## Results

### Theoretical calculations for phase engineering of Te

To understand the multiple controllable growth mechanisms in phase-engineered synthesis and the orientation of $\alpha$-Te nanosheets and $\beta$-Te nanoribbons under the synergistic effects of atomic cluster density and interface regulation, first-principles calculations based on density functional theory (DFT) were performed. At low flow rates, the lower atomic cluster density caused by the long distance between Te atoms

leads to fewer interactions among these atoms. In this case, the formation energy between Te atoms and the substrate is dominant. The formation energy ($E_f$) of $\alpha$-Te and $\beta$-Te on the WS$_2$ substrate were calculated by using the following equation:

$$E_f = \frac{E_{\text{total}} - nE_{\text{Te}} - E_{\text{sub}}}{n} \tag{1}$$

where $E_{\text{total}}$, $E_{\text{Te}}$, and $E_{\text{sub}}$ are the energies of the whole system, the isolated Te atom and the substrate, respectively. $n$ is the number of added Te atoms. As shown in Fig. 1a, compared to that of $\beta$-Te, one $\alpha$-Te cluster has a smaller formation energy with the WS$_2$ substrate (~−306.3 kJ mol$^{-1}$). This means that the Te atom more easily combines with the WS$_2$ substrate to form $\alpha$-Te, which then grows into single-layered crystals at low Ar flow rates.

At high flow rates, the atomic cluster density of Te atoms is greater, resulting in strong interactions. In this case, the formation energy of Te clusters plays a dominant role. Te atoms condense and form clusters before depositing on the substrate. Here, $E_{\text{sub}}$ in Eq. (1) is zero. The calculated results show that the formation energy of $\alpha$-Te is -2.6 kJ mol$^{-1}$ higher than that of $\beta$-Te (Fig. 1b), which indicates that the $\beta$-Te cluster preferentially formed and then bonded with WS$_2$ substrate. Finally, the $\beta$-Te clusters deposited on WS$_2$ surface continue to grow by adsorbing additional Te atoms at the edges to form $\beta$-Te single-layered crystals at high Ar flow rates.

To obtain a more thorough understanding of the phase transition process of $\alpha$-Te nanosheets and $\beta$-Te nanoribbons by means of an atomic cluster density and interface-guided multiple control strategy, we calculated the formation energy of Te atoms combined with the WS$_2$ substrate to form $\alpha$-Te nanosheets ($\alpha$-Te@WS$_2$) and the formation energy of Te atoms combined with each other to form $\beta$-Te clusters with respect to atomic cluster density, as shown in Fig. 1c. As the flow rate increases, the atomic cluster density increases accordingly, and the formation energy of the two modes decreases gradually. With a density lower than ~27%, the formation energy of $\alpha$-Te@WS$_2$ is lower than that of $\beta$-Te clusters, in which case $\alpha$-Te nanosheets are obtained. Then, with a density greater than ~27%, Te atoms will be formed into $\beta$-Te clusters. Moreover, the phase transition barrier between 2D $\alpha$-Te and $\beta$-Te is at least ~80 kJ/mol (Fig. 1d). The large energy barrier of the phase transition between $\alpha$-Te and $\beta$-Te during the growth process make it almost impossible for this transition to occur. Therefore, stable phases of $\alpha$-Te and $\beta$-Te were already formed during the nucleation stage.

There are two growth orientations of $\beta$-Te obtained by adsorbing additional Te atoms at the edges: along the armchair direction or the zigzag direction. The $E_f$ along these two directions was calculated by using the following equation:

$$E_f = \frac{E_{\text{total}} - E_{\text{sub+clu}} - nE_{\text{Te}}}{n} \tag{2}$$

where $E_{\text{sub+clu}}$ is the energy of the substrate with deposited clusters. The results indicate that the preferred growth direction of the $\beta$-Te clusters is along the armchair direction, which the formation energy is lower, as shown in Fig. 1e. The influence of interfacial interaction between Te atoms and the WS$_2$ substrate on the growth process was also investigated (Supplementary Fig. 2), which further suggests that $\beta$-Te is more inclined to form nanoribbons along the armchair direction with an angle of 60° between each ribbon.

On the other hand, the influence of the interface between different substrates and Te single crystals was investigated. The growth of Te on different substrates is the result of a dynamic balance between precursor adsorption and desorption and surface diffusion. DFT calculations revealed that the $E_f$ of growth for $\alpha$-Te on the WS$_2$ substrate is −306.3 kJ/mol, and the energy of growth on the SiO$_2$ (001) substrate is

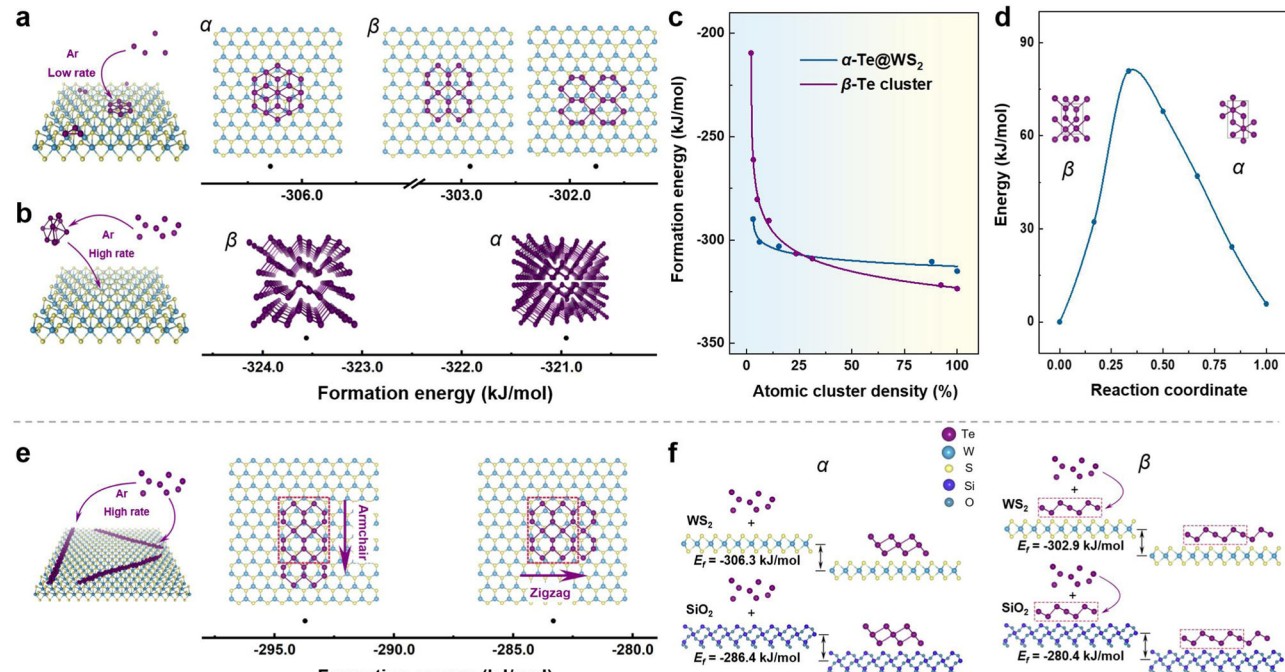

**Fig. 1 | Thermodynamic analysis of the phase engineering of ultrathin Te single crystals. a** Formation energies of $\alpha$-Te and $\beta$-Te on a WS$_2$ substrate. **b** Formation energies of $\alpha$-Te and $\beta$-Te in the bulk phase. **c** The formation energies of $\alpha$-Te@WS$_2$ and $\beta$-Te with respect to the atomic cluster density. The curves are fitted by the function "$y = A + B/(x + C)^D$", where $y$ is the formation energy, $x$ is the atomic cluster density, and A, B, C, and D are rational numbers. The curve obtained from this fitting closely aligns with the calculated points and serves to better illustrate the intersection point of the formation energies of two different growth modes.

**d** Calculated transition barrier between the $\alpha$ and $\beta$ phases. The insets are bilayer $\beta$-Te and the rectangular supercell of $\alpha$-Te. The blue points denote calculated energy values. **e** Formation energies of $\beta$-Te along the armchair and zigzag directions on the WS$_2$ substrate. Insets show the formation processes and atomic structures of $\alpha$-Te and $\beta$-Te. Atomic color code: purple, Te; blue, W; yellow, S. The black points in **a, b, e** denotes calculated formation energy values. **f** Calculated formation energies for $\alpha$-Te and $\beta$-Te on WS$_2$ and SiO$_2$ substrates, respectively.

−286.4 kJ/mol (Fig. 1f), which indicates that the WS$_2$ substrate is conducive to the horizontal growth of $\alpha$-Te nanosheets. Moreover, our calculations also showed that $\beta$-Te nanoribbons are likely to extend in the armchair direction on the SiO$_2$ substrate, and the formation energy is −280.4 kJ/mol, which is greater than that on the WS$_2$ substrate (−302.9 kJ/mol). Therefore, the $\beta$-Te nanoribbons are more inclined to extend in the plane on the WS$_2$ substrate so that it is easier to obtain thinner $\beta$-Te. This was verified by the corresponding experiment (Supplementary Fig. 3).

## Phase-engineered growth of $\alpha$-/$\beta$-Te on WS$_2$

Atomically thin $\alpha$-Te nanosheets and $\beta$-Te nanoribbons can be selectively synthesized by carefully regulating the growth parameters via the vdW epitaxy method and using monolayer WS$_2$ as a substrate (see details of the synthesis method in the Experimental Section). By systematically increasing the Ar flow rate, the ultrathin Te single crystals transform from $\alpha$-Te nanosheets to $\beta$-Te nanoribbons (as shown in Fig. 2a–c) with a uniform and tunable thickness, as distinguished by the thickness-dependent optical contrast (Fig. 2d–g and Supplementary Fig. 4) and representative atomic force microscopy (AFM) images (Supplementary Fig. 5). At a lower Ar flow rate of 80 sccm (Fig. 2d), $\alpha$-Te nanosheets are preferentially formed with a thickness of ~1.4-2 nm and average lateral edge sizes of ~2.2-3.1 μm. Upon increasing the Ar flow rate to 125 sccm (Fig. 2e), the resulting ultrathin Te single crystals exhibit a mixed phase with $\alpha$-Te nanosheets (with a thickness of ~0.4-0.8 nm and lateral dimensions of ~1.2-2 μm) and $\beta$-Te nanoribbons (with a thickness of ~0.4-0.8 nm and a length of ~10-16 μm). Upon further increasing the Ar flow rate from 150 to 175 sccm (Fig. 2e, f), the $\alpha$-Te nanosheets completely disappear, the thickness of the $\beta$-Te nanoribbons increases from ~1.3-3.2 nm to ~3.6-5 nm, and the length of the $\beta$-Te nanoribbons increases from ~12-24 μm to

~14-33 μm. The thicknesses of the Te samples prepared at different gas flow rates are measured statistically in Supplementary Fig. 4a–g. A summary of the evolution of the phase transition and thickness of ultrathin Te single crystals as a function of the Ar flow rate (50 to 200 sccm) is shown in Supplementary Fig. 4h. In particular, AFM images demonstrate that the highly uniform thicknesses of the $\alpha$-Te nanosheets and $\beta$-Te nanoribbons are approximately 0.4 nm (Supplementary Fig. 5a, i), indicating the formation of monolayer $\alpha$-Te nanosheets and $\beta$-Te nanoribbons. Notably, the $\alpha$-Te nanosheets are parallel to the WS$_2$ substrate, and the $\beta$-Te nanoribbons grow at three specific angles (30° and 90°) relative to WS$_2$, which may be attributed to the difference in the crystal symmetry orientations of the $\beta$-Te nanoribbons and WS$_2$[14].

## Structural characterization of $\alpha$-/$\beta$-Te on WS$_2$

To characterize the atomic structural morphology and composition of the $\alpha$-Te nanosheets and $\beta$-Te nanoribbons on monolayer WS$_2$, further characterizations were performed. The X-ray diffraction (XRD) patterns indicate that the crystal structures of $\alpha$-Te (Fig. 2h) and $\beta$-Te (Fig. 2i) belong to the $R$-$3$ $m$ (166) space group and $P3_121$ (152) space group, consistent with PDF#97-005-2499 and PDF#36-1452, respectively[13,34]. Moreover, all the $\alpha$-Te nanosheets and $\beta$-Te nanoribbons are well aligned in the [101] and [100] directions, respectively. The corresponding Raman spectra of $\alpha$-Te (Fig. 2j) and $\beta$-Te (Fig. 2k) exhibit three obvious peaks at 104 cm$^{-1}$ ($E_1$ mode), 112 cm$^{-1}$ ($A_1$ mode), 140 cm$^{-1}$ ($E_2$ mode) and 96 cm$^{-1}$ ($E_1$ mode), 126 cm$^{-1}$ ($A_1$ mode), 145 cm$^{-1}$ ($E_2$ mode), in accordance with previous studies[13,34]. In addition, the corresponding Raman mapping images of $\alpha$-Te and $\beta$-Te at 106 cm$^{-1}$ and 126 cm$^{-1}$ are given in the insets of Fig. 2j, k, respectively, exhibiting a rather uniform color contrast, indicating the highly uniform crystal structures of $\alpha$-Te and $\beta$-Te.

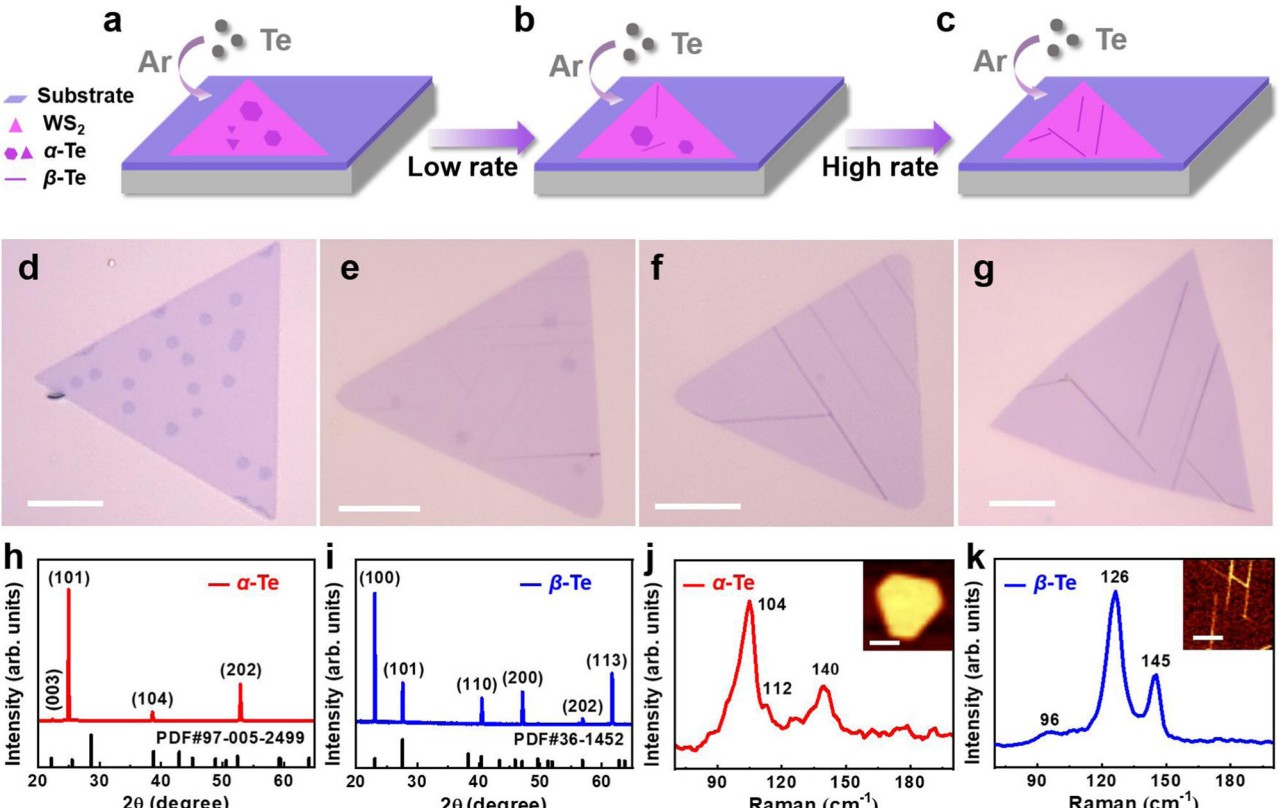

**Fig. 2 | Phase-engineered synthesis of α-/β-Te on monolayer WS₂. a–g** Schematic illustration (**a–c**) and optical microscopy images (**d–g**) of ultrathin Te single crystals on WS₂ as the Ar gas flow rate increased from 80 to 175 sccm at a growth temperature of 470 °C and a growth time of 10 min. **h, i** X-ray diffraction (XRD) characterization of α-Te nanosheets (**h**) and β-Te nanoribbons (**i**). The standard XRD cards PDF#97-005-2499 and PDF#36-1452 in **h** and **i** represent the *R-3 m* (166)

and *P3₁21* (152) space groups, corresponding to the α-Te and β-Te crystal structures, respectively. **j, k** Typical Raman spectra of α-Te nanosheets (**j**) and β-Te nanoribbons (**k**). The insets show the corresponding Raman mapping images of the α-Te nanosheets (**j**) and β-Te nanoribbons (**k**) at 104 and 126 cm⁻¹, respectively. Scale bars: 10 μm in (**d–g**) and 2 μm in (**j, k**).

According to the atomic-resolution high-angle annular dark field scanning transmission electron microscopy (HAADF-STEM) images, few point defects or dislocations are observed throughout the whole sample, indicating that the synthesized α-Te nanosheets and β-Te nanoribbons are single crystals with high quality and high crystallinity (Fig. 3 and Supplementary Fig. 6). Atomic-resolution HAADF-STEM image analysis revealed that the α-Te nanosheets exhibit a hexagonal 1 T phase, with each hexagonally arranged Te atom surrounded by six Te atoms, and the distance between the Te-Te atoms was 0.21 nm (Supplementary Fig. 6c). The β-Te nanoribbons exhibit helical chains and threefold screw symmetry, and the distance between Te-Te atoms is 0.32 nm (Supplementary Fig. 6f). Furthermore, based on the lattice constants of WS₂ (0.315 nm) and α-Te (0.420 nm), obvious moiré patterns with the smallest periodically repeating cell of ~1.26 nm are clearly observed in the α-Te/WS₂ heterostructure (3 × 3 α-Te unit cells stacked on 4 × 4 WS₂ unit cells), as shown in Fig. 3a, which is consistent with the simulated atomic structure model (inset). In contrast, the smallest periodically repeating cell of the β-Te/WS₂ heterostructures is ~5.34 nm (12 × 12 β-Te unit cells stacked on 17 × 17 WS₂ unit cells) due to the significant difference in lattice constants between β-Te and WS₂ (Fig. 3b).

Figure 3c–p show high-resolution transmission electron microscopy (HRTEM) images, corresponding selected area electron diffraction (SAED) patterns and energy dispersive energy dispersive spectrometer (EDS) elemental analysis results for α-Te nanosheets and β-Te nanoribbons epitaxially grown on monolayer WS₂. In Fig. 3c, e, HRTEM images of the α-Te/WS₂ region and β-Te/WS₂ region show well-resolved lattice spacings of ~0.398 nm and ~0.209 nm and ~0.193 nm

and ~0.225 nm, corresponding to the interplanar spacings of the (003) and (110) planes of the α-Te nanosheets and the (200) and (110) planes of the β-Te nanoribbons, respectively. The corresponding SAED pattern of the α-Te/WS₂ heterostructure shows two sets of hexagonally arranged diffraction spots (Fig. 3d), with calculated lattice spacings consistent with those of WS₂ (~0.273 nm, blue circles) and α-Te (~0.398 nm and ~0.209 nm, red circles). As shown in Fig. 3f, the SAED pattern of the β-Te/WS₂ heterostructure exhibits two sets of arranged diffraction spots with sixfold symmetry and orthogonal symmetry, with calculated lattice spacings consistent with those of WS₂ (~0.273 nm, blue circles) and β-Te (~0.193 nm and ~0.225 nm, green circles), respectively[13]. The EDS spectra in Fig. 3k, p show that, except for the Cu signal from the grid, only Te, W and Se signals were collected from the α-Te/WS₂ region and β-Te/WS₂ region, indicating the formation of the α-/β-Te/WS₂ heterostructure without impurities. In addition, the EDS elemental mapping images of S, W, and Te in the α-Te/WS₂ (Fig. 3h–j) and β-Te/WS₂ (Fig. 3m–o) heterostructures clearly show that Te, S, and W are uniformly distributed throughout these regions.

We further explored the effects of other parameters, such as growth temperature and growth time, on the growth behavior of α-Te nanosheets and β-Te nanoribbons under specific experimental conditions. As the growth temperature increases, the growth behavior largely transforms from kinetically controlled to thermodynamically controlled, leading to thicker and larger α-Te nanosheets (Supplementary Fig. 7) and β-Te nanoribbons (Supplementary Fig. 8). In addition, prolongation of the growth time leads to a higher Te partial pressure and a greater supply of precursor, resulting in thicker and

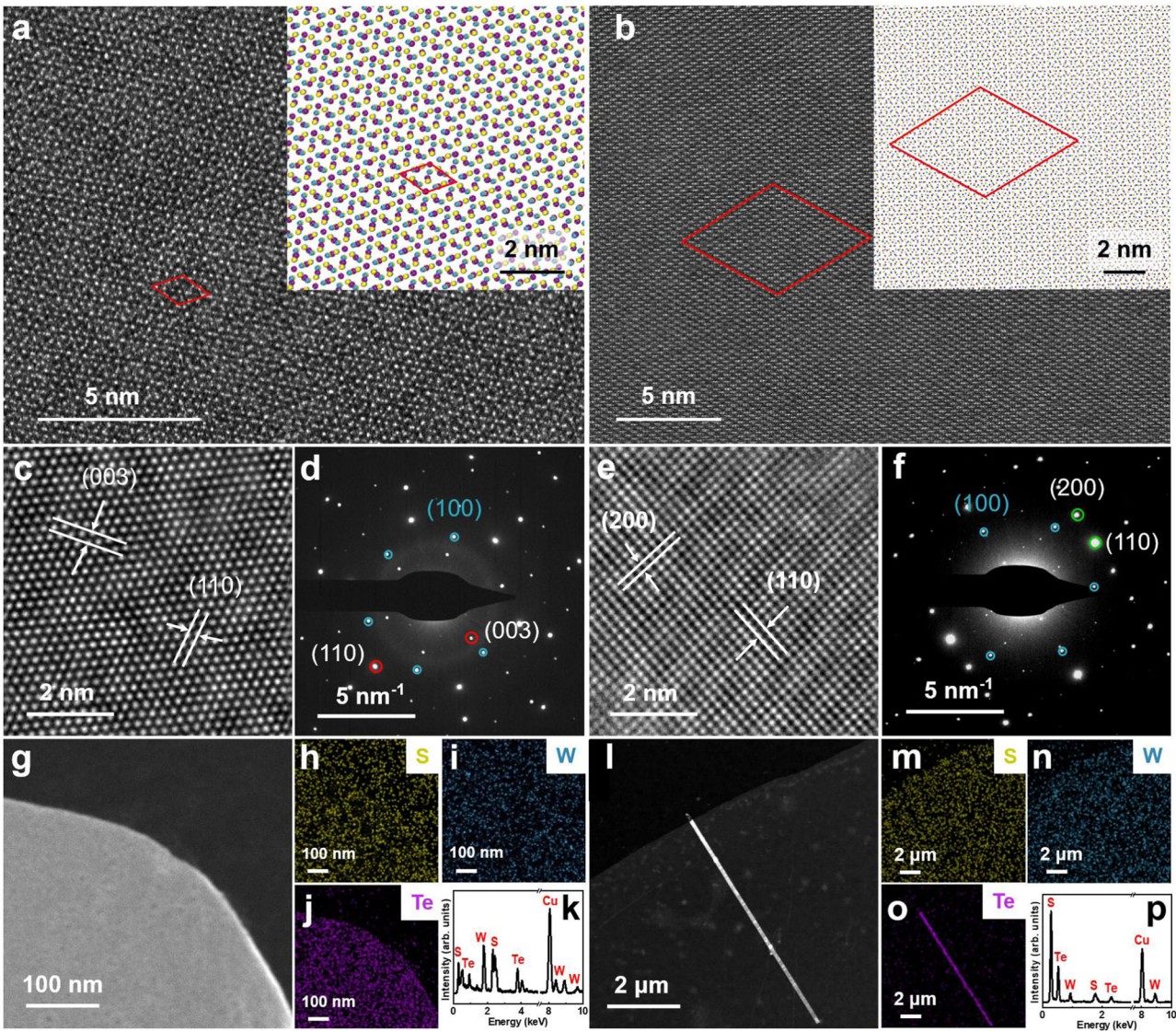

**Fig. 3 | Structural characterization of α-Te and β-Te on WS₂. a, b** Atomic-resolution scanning transmission electron microscopy (STEM) images of α-Te nanosheets (**a**) and β-Te nanoribbons (**b**) on WS₂. The insets show top views of atomic models of the α-Te/WS₂ (**a**) and β-Te/WS₂ (**b**) heterostructures. Atomic color code: purple, Te; blue, W; yellow, S. The red parallelograms in **a**, **b** and the corresponding insets highlight the periodically repeating moiré cells of α-Te/WS₂ (**a**) and β-Te/WS₂ (**b**), respectively. **c, e** High-resolution transmission electron microscopy (HRTEM) images of α-Te nanosheets (**c**) and β-Te nanoribbons (**e**) on WS₂.

**d, f** Selected area electron diffraction (SAED) patterns of the α-Te/WS₂ vdW crystal (**d**) and β-Te/WS₂ vdW crystal (**f**). The blue, red, and green circles in **d** and **f** represent the SAED patterns for WS₂, α-Te, and β-Te, respectively. **g, l** Low-magnification transmission electron microscopy images of α-Te nanosheets (**g**) and β-Te nanoribbons (**l**) on WS₂. **h–p** Energy dispersive spectrometer (EDS) mapping and EDS elemental analysis of α-Te nanosheets (**h–j, k**) and β-Te nanoribbons (**m–o, p**) on WS₂. Atom color code: purple, Te; blue, W; yellow, S.

larger α-Te nanosheets (Supplementary Fig. 9) and β-Te nanoribbons (Supplementary Fig. 10). More details of the growth behavior are described in the Supplementary Information.

### Electrical transport characteristics of 2D α/β-Te devices

This precise phase-control process provides a path for tailoring the band structure and thus electronic properties. The band structures and electronic properties of α-Te and β-Te are different due to differences in the atomic spatial configuration[4,35]. We analyzed the changes in the bandgaps of the α-Te (Fig. 4a) and β-Te (Fig. 4d) structures at different thicknesses according to first-principles calculations and density functional theory. The results of PBE functional calculation show that bulk α-Te clearly exhibits metallic properties, while bulk β-Te is an indirect bandgap semiconductor. As shown in Fig. 4a and Supplementary Fig. 11a, b, five-layer α-Te is still a metal with no bandgap, similar to the bulk phase. As the thickness of the α-Te

nanosheet decreases to four layers, α-Te begins to exhibit semiconducting behavior, and monolayer α-Te is a semiconductor with an indirect bandgap of 0.69 eV. As shown in Fig. 4d and Supplementary Fig. 11c, d, as the thickness of β-Te decreases from the bulk to the monolayer, β-Te is always a semiconductor, and the bandgap increases from 0.09 eV to 1.27 eV. Note that the bandgap of bulk β-Te calculated using Heyd-Scuseria-Ernzerhof functional[36] with the spin-orbit coupling method (HSE + SOC) is 0.34 eV, which is in good agreement with the previous experimental results[2]. The HSE + SOC calculation also indicates that bulk α-Te is a metal, which is consistent with PBE. Considering the burden of calculation and the similarity of band structure between PBE and HSE + SOC (Supplementary Fig. 12), we only use the PBE functional to investigate the influence of material thickness on the band structure of Te.

To experimentally examine the difference in electrical properties, we fabricated ultrathin α-Te nanosheet devices and β-Te nanoribbon

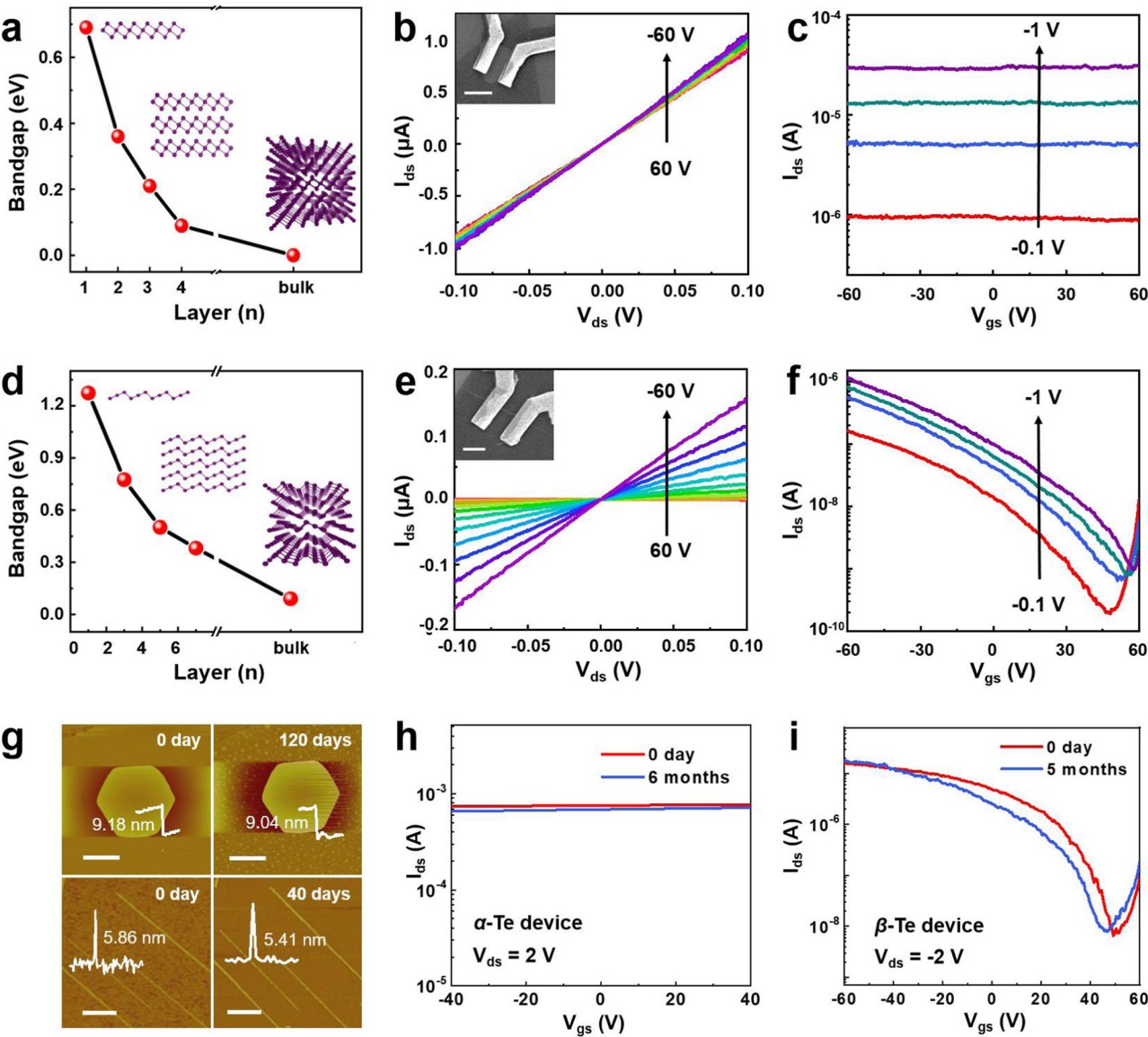

**Fig. 4 | Band structures and electrical characterization of $\alpha$-/$\beta$-Te transistors.** **a, d** Bandgaps of $\alpha$-Te nanosheets (**a**) and $\beta$-Te nanoribbons (**d**) as a function of the number of atomic layers. The red points denote calculated bandgap values, and the purple balls denote Te atoms. **b, c, e, f** The drain source current-drain source voltage·($I_{ds}$-$V_{ds}$) output curves under various gate voltages from 60 V to -60 V (−10 V step) and the drain source current-gate source voltage·($I_{ds}$-$V_{gs}$) transfer curves under various bias voltages (red, -0.1 V; blue, -0.4 V; green, -0.7 V; purple,

−1 V) for the $\alpha$-Te/WS$_2$ (**b**, **c**) and $\beta$-Te/WS$_2$ devices (**e**, **f**), respectively. The insets show optical images of the $\alpha$-Te/WS$_2$ (**b**) and $\beta$-Te/WS$_2$ (**e**) devices with thicknesses of 3.2 nm and 2.1 nm, respectively. Scale bars in (**b** and **e**): 2 μm. **g** AFM image of the $\alpha$-Te nanosheets and $\beta$-Te nanoribbons at different times. Scale bars: 2 μm. **h, i** Transfer curves of the $\alpha$-Te (**h**) and $\beta$-Te devices (**i**) at 0 (red line) and several months (blue line), respectively.

FETs in which Cr/Au metal electrode pairs were used as the source and drain electrodes and a Si substrate was used as the back gate. First, we investigated the thickness-dependent electrical characteristics of the $\alpha$-Te devices. The drain source current-drain source voltage ($I_{ds}$-$V_{ds}$) output characteristics of a 3.2 nm-thick $\alpha$-Te nanosheet device exhibit almost linear and symmetric properties, indicating the occurrence of ohmic contact between the metal electrodes and the $\alpha$-Te nanosheet (Fig. 4b). The drain source current-gate source voltage ($I_{ds}$-$V_{gs}$) transfer curves of this 3.2 nm-thick $\alpha$-Te nanosheet device are almost independent of the gate bias, indicating the typical metallic behavior of the $\alpha$-Te nanosheet (Fig. 4c). When the thickness of the $\alpha$-Te nanosheets is reduced to 2.2 nm, the output and transfer characteristics of these $\alpha$-Te nanosheets exhibit typical semimetal behavior (Supplementary Fig. 13c, d). As the thickness of the $\alpha$-Te nanosheets continues to decrease, the semimetallic properties gradually change to

semiconducting properties, and the ON/OFF ratio of the device gradually increases (Supplementary Fig. 13c–j). Furthermore, the $I_{ds}$-$V_{ds}$ and $I_{ds}$-$V_{gs}$ curves of an ~1.2 nm thick $\alpha$-Te device demonstrates typical n-type semiconductor behavior, with a calculated transconductance of $9 \times 10^{-7}$ S.

As shown in Fig. 4e, the $I_{ds}$-$V_{ds}$ curves of a 2.1 nm-thick $\beta$-Te transistor on a WS$_2$/285-nm SiO$_2$ substrate show linear characteristics. The transfer characteristics of the 2.1 nm-thick $\beta$-Te nanoribbon FET exhibit typical p-type behavior, indicating that holes dominate the charge transport process (Fig. 4f). Remarkably, devices exhibiting p-type-transistor behavior with a high ON-current density ($I_{ON}$) ~ 1000 μA μm$^{-1}$ are obtained on the WS$_2$/SiO$_2$ substrate at room temperature for $\beta$-Te transistors with a thickness of ~1.5-6.7 nm (Supplementary Fig. 14 and Supplementary Table 1). Among them, a maximum value of ~1200 μA μm$^{-1}$ is obtained for a 5 nm-thick $\beta$-Te nanoribbon transistor.

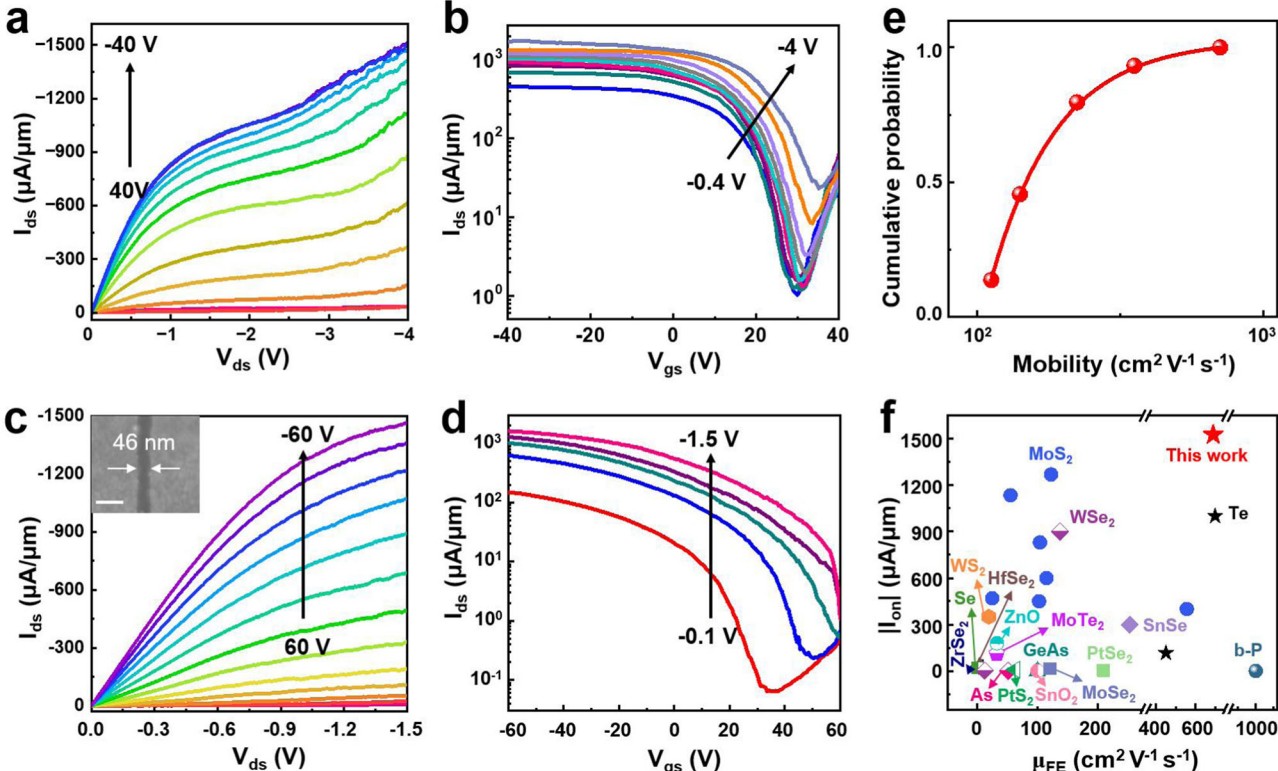

**Fig. 5 | Electrical characterization of the $\beta$-Te transistor. a, b** The drain source current-drain source voltage·($I_{ds}$-$V_{ds}$) output curves under various gate voltages from 40 V to -40 V (-5 V step) (**a**) and the drain source current-gate source voltage·($I_{ds}$-$V_{gs}$) transfer curves under various bias voltages from -0.4 (blue) to −1 V (purple) (-0.3 V step) and from −1 V (purple) to -4 (blue-gray) (-0.5 V step) (**b**) of the $\beta$-Te device with a channel length of 1.02 μm on h-BN/80-nm $Si_3N_4$/Si substrates. **c, d** Output curves under various gate voltages from 60 V to −60 V (−10 V step) (**c**) and transfer curves under various bias voltages (red, −0.1 V; blue, −0.4 V; green,

−0.7 V; purple, −1 V; pink, −1.5 V) (**d**) of the $\beta$-Te device with a channel length of -46 nm on h-BN/80-nm $Si_3N_4$/Si substrates. The scale bar in the inset **c**: 100 nm. **e** Cumulative distribution of the two-terminal mobility values of 44 $\beta$-Te transistors. The red dots and solid line represent the cumulative distribution values and trend of mobility, respectively. **f** Comparison of the ON-state current density and mobility with those of other 2D materials[2,8,28,29,38–56]. The detailed values of (**f**) are shown in Supplementary Table 3.

Moreover, the electrical properties are mainly derived from 2D Te single crystals but not from the $WS_2$ substrate (Supplementary Figs. 15, 16).

In addition, the 2D Te synthesized on the $WS_2$ substrate exhibited good environmental and electrical stability after several months (Fig. 4g–i and Supplementary Fig. 17). As shown in Fig. 4g, the AFM studies reveal that $\alpha$-Te nanosheets or $\beta$-Te nanoribbons do not show significant changes in either morphology or thickness after months of exposure to air, suggesting that they are highly air-stable. Furthermore, we studied the electrical stability of the $\alpha$-Te and $\beta$-Te device. Figure 4h, i and Supplementary Fig. 17 show the transfer characteristics of the $\alpha$-Te and $\beta$-Te devices, which were measured immediately and after several months, respectively. No significant degradation was observed in either the $\alpha$-Te or $\beta$-Te devices during the 5-6 months period (Fig. 4h, i), which proves the good electrical stability of the metallic $\alpha$-Te device and semiconducting $\beta$-Te device.

**Prominent p-type electrical properties of 2D $\beta$-Te transistors**
The current of the device increases with increasing gate dielectric capacitance ($C_g = \varepsilon_{ox}\varepsilon_O S/d$), where $\varepsilon_{ox}$ and $d$ are the oxide dielectric constant and dielectric thickness, respectively. To achieve the same current density with a larger $C_g$, the gate voltage can be decreased, which means that a thinner high-$k$ dielectric gate can better control the device. Therefore, we expect to determine the intrinsic electronic properties of $\beta$-Te transistors and minimize the influence of the substrate using thinner high-$k$ gate dielectrics. Moreover, the interface quality can improve to high $k$ through dielectrics to reduce substrate

phonon scattering and charge impurities[13]. Obviously, the electrical output and transport studies at room temperature show that the -5.2 nm- and -14.3 nm- $\beta$-Te nanoribbon transistors with thinner gate dielectrics (h-BN/80-nm $Si_3N_4$/Si) exhibit consistent p-type transistor behaviors and higher ON-state current densities of up to -1300 μA μm$^{-1}$ and -1500 μA μm$^{-1}$, respectively. (Fig. 5a, b and Supplementary Fig. 14m–p).

To achieve high $I_{ON}$ values for the $\beta$-Te transistor at a $V_{ds}$ of approximately 1 V or less, the channel length is scaled to the sub-50 nm regime without compromising the channel or contact performance (Fig. 5c). More importantly, the $I_{ON}$ of the 8 nm-thick $\beta$-Te transistor readily reaches 1270 μA μm$^{-1}$ at $V_{ds}$ = −1 V, as evidenced by both the output and transfer characteristics, with a lower ON-state resistance ($R_{on}$) of 0.63 kΩ μm (for consistency, $R_{on}$ is evaluated by V/I in the linear regime before current saturation). The ON/OFF ratio at a small $V_{ds}$ = −0.01 V is $2.4 \times 10^3$, which is still a reasonable value considering its narrow bandgap of -0.3 eV. As $V_{ds}$ increases, the ON/OFF ratio deteriorates, and the threshold voltage shows a positive shift, which may be attributed to the competitive control of the transistor channel by the gate voltage and drain-source bias in short-channel devices. Under a high drain bias, the electric field of the drain overpowers that of the gate, compromising the controllability of the gate[37].

Moreover, Fig. 5e demonstrates that the cumulative hole mobility distribution of 44 $\beta$-Te nanoribbon FETs is more than 100 cm$^2$ V$^{-1}$ s$^{-1}$, with a maximum mobility of 690.7 cm$^2$ V$^{-1}$ s$^{-1}$, which is consistent with our theoretical calculations that revealed that the hole mobility can reach 1134 (armchair) and 1346 cm$^2$ V$^{-1}$ s$^{-1}$ (zigzag) (Supplementary

Table 2). Obviously, the ON-state current density (1527 μA μm⁻¹) and mobility (690.7 cm² V⁻¹ s⁻¹) of the β-Te nanoribbon FETs are greater than those of most 2D single-crystal FET devices, such as those based on Te[2,8], WSe$_2$[38,39], PtSe$_2$[40], ZrSe$_2$[41], HfSe$_2$[41], Se[42], WS$_2$[43], ZnO[44], As[45], MoS$_2$[28,29,43,46–49], MoTe$_2$[50], PtS$_2$[51], SnO$_2$[52], GeAs[53], SnSe[54], b-P[55], and MoSe$_2$[56] (Fig. 5f and Supplementary Table 3).

Remarkably, to construct a large-area Te single crystal and fabricate and integrate 2D Te into electronic and optoelectronic devices, we prepared a high-performance 2D Te nanoribbon on highly oriented monolayer 2-inch MoS$_2$ films (Supplementary Fig. 18a–c). The strong single-orientation (90%) epitaxy of β-Te nanoribbons with a density, size and thickness of approximately 15,000 pieces/mm², 5-12 μm and 0.8-5.6 nm on monolayer MoS$_2$ is schematically shown in Supplementary Fig. 18, which provides a reliable path for the future preparation of large-area, high-quality, and scalable 2D Te arrays and single-crystal films, laying a solid foundation for the future integration and industrial application of 2D Te devices.

## Discussion

In summary, the phase transition processes, geometry, and thickness of α-Te nanosheets and β-Te nanoribbons are precisely controlled in terms of the synergistic effects of atomic cluster density and interface regulation. The α-Te nanosheets and β-Te nanoribbons were further verified by XRD, Raman spectroscopy, and STEM characterization. Electrical transport studies revealed that the α-Te nanosheets exhibit semimetallic behavior with a thickness-tunable bandgap, while the 2D β-Te nanoribbons exhibit excellent p-type semiconducting properties (the mobility of ~690.7 cm² V⁻¹ s⁻¹, I$_{ON}$ of ~1527 μA μm⁻¹). Both α-Te nanosheets and β-Te nanoribbons exhibit excellent air stability after several months. By employing the continuous vdW epitaxy growth of different phases of Te, achieving heteroepitaxial integration of metal-semiconductor states within the same atomic plane has become feasible, enabling atomic-scale metal-semiconductor contacts and reducing contact resistance. Therefore, the realization of Te phase engineering would pave the way for enhancing the performance of optoelectronic devices, wearable electronics, synaptic devices, and friction-based nanogenerators, laying the groundwork for large-scale, high-performance integrated circuits.

## Methods
### CVD growth of WS$_2$ monolayers
WS$_2$ monolayers were fabricated on SiO$_2$/Si substrates by an atmospheric pressure CVD process. First, sodium chloride (NaCl) was ground with an onyx mortar and pestle to a powder particle size of approximately 200 mesh, and then, ground sodium chloride (NaCl) was homogeneously mixed with tungsten trioxide powder (99.8%, Aladdin) at a rate of 0.05 g: 0.5 g. The mixed powder and a small amount of sulfur powder (99.5%, Alfa Aesar) were then placed in the highest temperature zone and upstream zone of the furnace, respectively. Then, a monolayer WS$_2$ was prepared at a flow rate of 200 sccm of Ar (99.995%), a growth temperature of 800 °C, and a growth time of 10 min.

### Mechanical stripping multilayer h-BN
The high-quality h-BN crystals were used as the source, mechanically peeled off with blue tape, repeated approximately 5 times, and then pasted on SiO$_2$/Si$_3$N$_4$ substrate, pressed the blue tape by finger for approximately 2 min and then tore it off, and screened the h-BN nanosheets with the size larger than 10 μm and the thickness less than 20 nm as the growth substrate.

### CVD growth of ultrathin α-/β-Te single crystals on WS$_2$/h-BN
First, the growth substrate, either a single layer of WS$_2$ obtained by one-step CVD preparation or a few layers of h-BN prepared by mechanical stripping, was placed downstream of a tube furnace with a uniform cooling gradient. A quartz boat containing Te powder (99.99%, Meryer) was then placed in the center of the tube furnace. The α-Te nanosheets on WS$_2$/h-BN were obtained by heating the furnace to 430–550 °C (with a cooling zone of 300–450 °C) for 2–20 min at a preset argon flow of 50–125 sccm. The corresponding β-Te nanoribbons on WS$_2$/h-BN can be obtained by heating the furnace to 430–550 °C (with the cooling zone at 250–450 °C) for 2–20 minutes at an argon flow rate of 125–200 sccm and a growth temperature of 310–510 °C (with the cooling zone at 250–410 °C) for 2–14 minutes.

### Material characterization
The morphologies of the α-Te/WS$_2$ and β-Te/WS$_2$ crystals were characterized using optical microscopy (OLYMPUS BX41 M-LED) and AFM (Bruker Dimension ICON). The crystal structures and phase purities were characterized via XRD (Smartlab3) and Raman spectroscopy (Witec Alpha 300 R confocal Raman microscope with a 532 nm laser, RT). The micromorphology was analyzed by STEM (Titan Cubed Themis G2300) and TEM (Talos F200X) operating at 200 kV and equipped with an EDS system.

### Device fabrication and characterization
Polymethyl methacrylate (PMMA) was spin-coated on SiO$_2$/Si wafers with α-Te on WS$_2$/ h-BN and β-Te on WS$_2$/ h-BN. Then, α-Te on WS$_2$/h-BN and β-Te on WS$_2$/ h-BN FETs were fabricated by electron beam lithography (EBL), in which Cr/Au with a thickness of 10 nm/50 nm was used for the contact electrodes. Electrical performance was assessed with a Lake Shore TTPX Probe Station and an Agilent 1500 A semiconductor device analyzer under vacuum at room temperature.

### Mobility calculations
The field-effect mobility of the β-Te devices was estimated using the equation[52]:

$$g_m = \frac{\mu_{fe} C_g}{L^2} V_{ds} \qquad (3)$$

Here, $g_m = dI_{ds}/dV_{ds}$ is the linear-region transconductance, $L$ is the channel length of the β-Te nanoribbons, and the capacitance coefficient $C_g$ can be calculated using the following equation:

$$C_g \frac{2\pi\varepsilon_r\varepsilon_0}{\cosh^{-1}(\frac{2h+d}{d})} L \qquad (4)$$

where $\varepsilon_r$ and $\varepsilon_O$ represent the relative dielectric constant and vacuum dielectric constant, respectively. $h$ is the thickness of the dielectric layer, and $d$ is the lateral size of the β-Te nanoribbons.

### DFT calculations
All calculations were performed within the framework of density functional theory (DFT) implemented in the Vienna ab initio simulation package (VASP)[57]. The electron-electron interactions were treated by a general gradient approximation parametrized by Perdew-Burke-Ernzerhof (PBE)[58] with van der Waals (vdW) correction (DFT-D3)[59]. A kinetic energy cutoff of 500 eV was set for the plane-wave expansion to ensure convergence[60]. A vacuum space of more than 20 Å was introduced in the perpendicular plane to avoid interlayer interactions. Electronic minimization was performed with a tolerance of 10⁻⁶ eV, and ionic relaxation was performed with a force tolerance of 10⁻² eV Å⁻¹ for each ion.

## Data availability
Relevant data supporting the key findings of this study are available within the article and the Supplementary Information file. All raw data generated during the current study are available from the corresponding authors upon request.

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

## Acknowledgements

J.P.L. acknowledges the National Key Research and Development Program of China (Grant No. 2023YFB3611400) and the National Natural Science Foundation of China (Grant No. 62174026). Z.H.N. acknowledges the National Natural Science Foundation of China (Grant Nos. 62225404, 61927808, and T2321002) and the Natural Science Foundation of Jiangsu Province, Major Project (BK20222007). B.Z. acknowledges the National Natural Science Foundation of China (Grant No. 62205055) and the Natural Science Foundation of Jiangsu Province (BK20220860).

## Author contributions

Z.-H.N., B.Z. and J.P.L. conceived the research and designed the experiments. J.Z. performed the experiments and experimental analysis. G.-T.Z. and Q.C. provided theoretical support. B.Z., J.Z., and W.H.W. carried out the device fabrication and electrical measurements. W.W.Z. and H.W.L. contributed to discussions and data analysis. Z.-H.N., B.Z., H.W.L., J.P.L. and J.Z. wrote the paper. All authors discussed the results and commented on the manuscript

## Competing interests

The authors declare no competing interests.
