## [Peer Review File · Nature Communications]

Phase-Engineered Synthesis of Atomically Thin Te Single Crystals with High ON-State CurrentsEditorial Note: Parts of this Peer Review File have been redacted as indicated to remove third-party material where no permission to publish could be obtained.

REVIEWER COMMENTS

Reviewer #1 (Remarks to the Author):

In the manuscript titled “Phase-Engineered Synthesis of Atomically Thin Te Single Crystals with High ON-State Currents”, the authors J. Zhou et al reported a detailed study on vdW epitaxial growth of alpha and beta-Te nanostructures in transition metal dichalcogenide templates, with the phase and morphology tunable with growth conditions. Credits should be given to authors to expand the knowledge of Te in certain aspects, eg. metal-to-semiconductor in alpha Te with reduced thickness, and excellent electrical performance of beta Te. However, I feel that overall, these merits do not warrant its publication in Nature Communications for the following reasons: (1) A similar vdW epitaxy growth of Te, and similar electrical performance has been reported (ACS Nano 2014, 8, 7, 7497); (2) the methodology of extracting field effect mobility and current density is likely flawed. Therefore, I do not recommend publication in NC given the high standard of the journal. In the case of re-submitting or transferring the manuscript, I would suggest the authors to consider addressing the formation of a single atomic chain of beta-Te as shown in Supplementary Fig 4i. This is, to my best knowledge, the first time a single atomic chain of bare Te is synthesized without encapsulation (Nature Electronics 3 (3), 141, 2020). The following are some major concerns on the manuscript:

1. On Page 8 Ln 13, the authors claimed highly uniform thickness for CVD grown thickness. Given the distribution of process-dependent thickness distribution provided in the same paragraph, “highly uniform” seems to be over-claiming. In addition, the authors need to provide statistics on how many flakes/nanoribbons are counted to provide this geometry range. Similarly, the error bars in Supplementary Fig. 3h need to be defined with the number of flakes measured.
2. On Pg9 paragraph 2: “consistent with PDF#97-005-2499 and PDF#36-1452” What does “PDF#xxx” mean?
3. The unit cell of beta Te in Fig.1 and Supplementary 1b is inconsistent with the structure reported in other literatures and the symmetry group. Beta Te has a 1D van der Waals structure with three-fold screw symmetry. Each tellurium atom forms 2 covalent bonds with 2 neighboring Te atoms within the chain (e. g. Nano Lett. 2017, 17, 6, 3965). However, in the manuscript, there is an additional interchain bonding, which makes three Te atoms inequivalent, and the crystal no longer belongs to P3121 symmetry group.
4. Pg 14, the calculated bulk bandgap of beta Te is 0.09 eV which is significantly lower than that reported in the literature (0.3-0.35 eV, e.g., ACS Nano 2018, 12, 7, 7253). Please clarify.
5. Pg 15: The authors extracted an electron mobility of 74.7 cm²/Vs in alpha Te. However there is only a small on/off ratio with the gate. While the semiconducting or semi-metallic behavior in thin alpha Te is somewhat interesting, it should be noted that it is not a technically a transistor and thus using the terminologies in FETs seems out of place. The field-effect mobility should be extracted in a region where dI/dV_g is constant over a sufficiently large window of V_g . In addition, a low V_d should be used to ensure the device is in the linear regime.
6. Multiple device electrical characteristics were displayed in the manuscript and Supplementary. I recommend the authors to add key device parameters such as thickness, W , and L to each figure, or number the devices so that the readers can easily follow.

7. Page 15: The current density in beta Te is certainly impressive. However, a potential pitfall is the way the authors normalize the current with width. Can the authors provide the information of the channel width W for these devices and how they are measured? Assuming a cylindrical morphology of these nanoribbons, the diameter would be only a few nm, which is hard to accurately measure. More importantly, in Eq (4) the authors used a model to calculate the capacitors where an effective channel width is defined by a hyperbolic equation, and it can lead to over-estimation of current density and mobility.

8. Following Q7, Eq (4) is typically used to calculate the parasitic capacitance of a cylindrical wire over a ground plane in a uniform dielectric. I am skeptical whether this is applicable to this device geometry since the nanowire is in air whereas the gate is isolated by SiO₂. A more rigid validation of capacitance calculation needs to be provided since it is critical to back up the key claim of superior electrical performance of p-type beta Te.

9. WS₂ should be semiconducting (see Nat. Commun. 12, 693, 2021) while in Supplementary Fig. 14, the WS₂ template is completely insulating. Why is that?

10. Pg. 16 the authors inferred the on-state current increase after 5 months is due to measurement error. Can the authors elaborate what measurement errors?

11. Pg. 18 Is the field effect mobility extracted from short-channel devices? It should be noted that in short-channel devices with a universal back gate, the field-effect mobility is often over-estimated because the gate is also modulating contacts. The field-effect mobility should only be extracted in long channel devices where the channel resistance is significantly larger than the contact resistance. The authors should report mobility with long channel devices where channel resistance is sufficiently larger than contact resistance, or may just report transconductance instead.

Reviewer #2 (Remarks to the Author):

This manuscript demonstrated phase engineering of tellurium and also showed some high performance devices made out of the Te. The authors realized different phases of Te by controlling the flow rate, which was not reported before, and showed systematically evolutionary study of this process. However, the importance of this work is not clearly manifested, the structure of this manuscript is not well organized, and is not of scientific rigorous. I think this manuscript should at least have a major revise or try other specific journals which focus on growth.

1. Although the manuscript showed realization of the phase engineering of Te, it didn't demonstrate the application of the different phases, especially the application of phase transition. So, the question raises -- what is the significance of realizing different phases (refer to MoTe₂)?

2. Te is of particularly interesting because it could be used for p-type high performance transistors, especially for back-end-of-line integration. The devices in this manuscript did not achieve particularly superior performance for this purpose. The mobility was not as high as previously reported (>700 cm²/V-s, Nature Electronics 1, 228(2018), Advanced Materials 30, 1803109 (2018), Mater. Today 63, 50-58 (2023)), and it did not achieve a particularly superior improvement in leakage current nor a fully back-end-of-line compatible preparation temperature, as demonstrated in paper (Applied Surface Science

636, 157801 (2023))

3. The author also did not introduce some of the most recent and important Te research results in the introduction. There have been many reports on the high quality growth of Te using different methods and realization of different phases and the study of devices, not limited to the reference mentioned above.

4. On page 3, paragraph 2, the authors wrote that vdW epitaxial growth is a newly developed technology, but in fact this technology has been reported at least in the 1990s. In addition, the authors mentioned vdW epitaxy many times in the manuscript, however, based on the growth structures, the growth method employed should belong to the concept of remote epitaxy.

5. In the third paragraph on page 3, the author discussed α -Te and β -Te without any introduction, so what is the difference between them? The manuscript did not explain in the following contents either.

6. Lack of basic information related to this research. Such as the structure of the substrate, the growth method of Te, because there are many ways to grow Te, solution based, MBE, and CVD... The CVD growth kinetics is very different from the other methods.

7. Although the authors stated that WS₂ is insulating and does not affect the conduction of Te devices, the authors also mentioned that there is a spatial charge transfer between the two. So a concern arises that whether the performance of Te achieved in this report is the intrinsic performance of Te or it depends on the specific substrates.

8. Following the above point, one of another biggest questions in the manuscript is whether the Te structures are a pure phase since the Te grows on the WS₂ substrate at a high temperature up to 550 C. Such a high temperature is likely to cause reaction between the substrate and the Te. At least it can be seen from Figure S13b on page 43 that W 4f_{5/2} to W 4f_{7/2} ratio is significantly changed indicating the WS₂ severely changed its chemistry, and it is not a pure phase anymore.

9. Once again, the manuscript lacks introduction to many details. On page 17, it mentioned the h-bn related dielectrics, but how was this realized?

Reviewer #3 (Remarks to the Author):

In this work, Zhou et al. realized the phase-engineered synthesis of α - and β -Te single crystals with controlled thicknesses, by delicately modulating the competition between interfacial interaction and cluster interplay. The intricate mechanisms underlying the phase engineering were elucidated through compelling theoretical calculations. The as-synthesized Te, particularly the β -Te, exhibited relatively good Field Effect Transistor (FET) performances. The presented data is of high quality, and the article is well-organized. However, I hesitate to recommend its publication in the journal of Nature Communications due to concerns regarding novelty and practical value. The detailed reasons are listed below:

1. The fabrication of β -Te has been frequently reported in recent years through various growth methods on versatile substrates (Nat. Electron. 1, 228-236 (2018), Adv. Mater. 2018, 30, 1803109 (2018), Nano-Micro Lett. 14, 109 (2022), Mater. Today 63, 50-58 (2023)). Some studies have achieved the direct synthesis of β -Te on Si/SiO₂ substrates (Adv. Mater. 2018, 30, 1803109 (2018)) or dielectric materials (e.g., hBN, Nano-Micro Lett. 14, 109 (2022)), making the fabrication process more industrially compatible. Moreover, certain reports on the FET performances of β -Te, including mobility and on-off

ratio, surpass those presented in this work. The growth methods, substrates, and FET performances in this study do not showcase significant advantages over these prior reports.

2. Although the authors successfully realize the phase engineering of α - and β -Te using WS₂ as a growth substrate, the potential applications or intriguing properties of α -Te seem limited. The synthesis of α -Te itself has also been previously reported. The significance of this phase engineering strategy warrants further consideration.

Besides, there some other issues that need to be addressed to enhance the overall quality of this work:

1. In the last sentence of page 3, the authors stated that synthesis of single-layer β -Te had not yet been achieved. However, based on my knowledge, this has been realized by Huang et al using an MBE route (Nano Lett. 17, 4619–4623 (2017)) and Wang et al. via a solution process (Nat. Electron. 1, 228-236 (2018)). It is advisable for the authors to scrutinize the accuracy of this statement.

2. In Figure 1c, the authors compared the formation energies of α - and β -Te, by calculating the formation energy of Te atoms combined with WS₂ substrate to form α -Te nanosheets (α -Te@WS₂) and the formation energy of Te atoms combined with each other to form β -Te clusters with respect to atomic cluster. Regarding β -Te, the influence of interfacial interaction between Te atoms and the WS₂ substrate on the growth process should also be considered. However, it appears that the authors did not account for this factor in their calculations.

3. The authors solely conducted AFM studies to substantiate the stability of α -Te. Additional characterizations may be warranted, particularly given the typical poor air stability associated with metallic 2D materials.

4. The last sentence of page 7 was repeated twice.

Point-by-point responses to the reviewers' comments on the manuscript "*Phase-Engineered Synthesis of Atomically Thin Te Single Crystals with High ON-State Currents*" (Manuscript ID: NCOMMS-23-48826).

We have carefully studied the comments and thoroughly made the necessary revisions. The point-by-point responses to the reviewers' comments are listed as follows:

Reviewers' comments and our responses

Reviewer #1:

In the manuscript titled "Phase-Engineered Synthesis of Atomically Thin Te Single Crystals with High ON-State Currents", the authors J. Zhou et al reported a detailed study on vdW epitaxial growth of alpha and beta-Te nanostructures in transition metal dichalcogenide templates, with the phase and morphology tunable with growth conditions. Credits should be given to authors to expand the knowledge of Te in certain aspects, eg. metal-to-semiconductor in alpha Te with reduced thickness, and excellent electrical performance of beta Te. However, I feel that overall, these merits do not warrant its publication in Nature Communications for the following reasons: (1) A similar vdW epitaxy growth of Te, and similar electrical performance has been reported (ACS Nano 2014, 8, 7, 7497); (2) the methodology of extracting field effect mobility and current density is likely flawed. Therefore, I do not recommend publication in NC

given the high standard of the journal. In the case of re-submitting or transferring the manuscript, I would suggest the authors to consider addressing the formation of a single atomic chain of beta-Te as shown in Supplementary Fig 4i. This is, to my best knowledge, the first time a single atomic chain of bare Te is synthesized without encapsulation (*Nature Electronics* 3 (3), 141, 2020). The following are some major concerns on the manuscript:

RESPONSE: We are grateful for the valuable time of the reviewer in reviewing our manuscript with positive comments such as “Credits should be given to authors to expand the knowledge of Te in certain aspects, eg. metal-to-semiconductor in alpha Te with reduced thickness and excellent electrical performance of beta Te.” “To my best knowledge, the first time a single atomic chain of bare Te is synthesized without encapsulation.” We also thank the reviewer for the in-depth analysis of our manuscript and for the many insightful comments. Below, we respond to each of the reviewers’ comments and identify specific revisions to our manuscript and the Supplementary Information.

1. Compared to the reported article (*ACS Nano* 2014, **8**, 7, 7497), the innovative aspects of our manuscript are as follows: (1) Although He et al. obtained 2D hexagonal Te nanoplates by vdW epitaxial growth, the thickness of the nanoplate was too large (~30-80 nm). In contrast, we successfully synthesized monolayer (~0.4 nm thick) α -Te nanosheets and β -Te nanoribbons by precisely controlling the binding/formation energy induced by the atomic cluster density and an interface-guided multiple control strategy. (2) The reported article primarily focused on the influence of chemically inert mica surfaces on the growth behavior of hexagonal Te nanoplates. However, there has been a lack of in-depth investigations into the electrical properties of Te, with only the response time and recovery time of Te nanoplates being reported to be 4.4 and 2.8 s, respectively. In contrast, our manuscript provides a comprehensive exploration of the outstanding and stable electrical properties of Te. For instance, β -Te nanoribbons are p-type semiconductors with hole mobilities as high as $\sim 690.7 \text{ cm}^2 \text{ V}^{-1} \text{ s}^{-1}$ and a high I_{on} of $1.27 \text{ mA } \mu\text{m}^{-1}$ at $V_{\text{ds}} = -1 \text{ V}$ at room temperature, and the α -Te nanosheets and β -Te nanoribbons maintain excellent electrical air stability after 5-6 months. (3) We

utilized phase engineering for the first time to achieve controlled synthesis of α -Te nanosheets and β -Te nanoribbons, delving into the distinct nucleation mechanisms and growth behaviors of the two phases. Additionally, we investigated the differing electrical properties of the two phases; for example, the α -Te nanosheet is a semimetal and undergoes a transition from a metal to an n-type semiconductor, whereas the β -Te nanoribbons consistently maintain their status as p-type semiconductors even as their thickness decreases. (4) We can grow large-area, highly oriented Te single crystals. The highly single-orientation (90%) epitaxy of β -Te nanoribbons on monolayer 2-inch MoS₂ provides a reliable path for the future preparation of large-area, high-quality, and scalable 2D Te arrays and single-crystal films. This has laid a solid foundation for the future integration and industrial application of 2D Te devices.

2. Regarding the extraction of the current density, we provided a detailed explanation in question 7. We divided the device current by the channel width (W), and this extraction method was valid and consistent with previous literature reports (*Nat. Electron.* 2018, **1**, 228-236; *Nature* 2023, **613**, 274-279; *Nature* 2023, **616**, 470-475). To ensure the accuracy of W, we conducted at least five measurements and obtained the average value.

Using the formula (Eq. 1) for β -Te nanoribbons reported in the literature (*Nat. Electron.* 2018, **1**, 228-236; *Adv Mater.* 2018, **30**, 1803109; *Mater. Today* 2023, **63**, 50-58) to calculate the mobility of β -Te nanoribbons in our article, the maximum value reaches 3752.27 cm² V⁻¹ s⁻¹. As our β -Te width is less than the dielectric layer thickness, to ensure the accuracy of the electrical performance and avoid overstating device capabilities, we utilized the formulas (Eq. 4 in our manuscript) (*Nano Lett.* 2009, **9**, 360-365; *Nature* 2021, **591**, 385-389; *Nano Lett.* 2007, **7**, 2463-2469) in this manuscript to calculate a mobility of ~690.7 cm² V⁻¹ s⁻¹.

$$\mu_{FE} = \frac{g_m L}{W \times C_g \times V_{ds}} \quad (\text{Eq. 1})$$

$$g_m = \frac{\mu_{fe} C_g}{L^2} V_{ds} \quad (\text{Eq. 4})$$

1. On Page 8 Ln 13, the authors claimed highly uniform thickness for CVD grown thickness. Given the distribution of process-dependent thickness distribution provided in the same paragraph, “highly uniform” seems to be over-claiming. In addition, the authors need to provide statistics on how many flakes/nanoribbons are counted to provide this geometry range. Similarly, the error bars in Supplementary Fig. 3h need to be defined with the number of flakes measured.

RESPONSE and CHANGES: Thanks for your careful reading and suggestions. We have revised the description of the uniform thickness in the manuscript to present the facts more rigorously (highlighted in red, page. 9, line 174). In addition, we have added statistical data on the thickness distribution of the synthesized samples at different flow rates (50-200 sccm) for a total of 225 samples, as shown in Supplementary Fig. 4 of the revised manuscript. Here, the thicknesses of different samples prepared at different flow rates were counted using the average of five measurements for each sample to ensure that the error values of the measurement results were minimized. In addition, all the error bars in the Supplementary Fig. 4h (the original Supplementary Fig. 3h) have been added.

At a lower Ar flow rate of 80 sccm (Fig. R1b), α -Te nanosheets are preferentially formed with a thickness of $\sim 1.7 \pm 0.3$ nm. Upon increasing the Ar flow rate to 125 sccm (Fig. R1d), the resulting ultrathin Te single crystals exhibit a mixed phase with α -Te nanosheets (with a thickness of $\sim 0.6 \pm 0.2$ nm) and β -Te nanoribbons (with a thickness of $\sim 0.6 \pm 0.2$ nm). Upon further increasing the Ar flow rate from 150 to 175 sccm (Fig. R1e-f), the α -Te nanosheets completely disappear, and the thickness of the β -Te nanoribbons increases from $\sim 2.25 \pm 0.95$ nm to $\sim 4.3 \pm 0.7$ nm. A summary of the evolution of the phase transition and thickness of ultrathin Te single crystals as a function of the Ar flow rate (50 to 200 sccm) is shown in Supplementary Fig. 4h.

We have added this statistical information on the thicknesses of the α -Te nanosheets and β -Te nanoribbons obtained from our growth in Supplementary Fig. 4 of the revised manuscript.

Fig. R1 Thickness characterization of synthesized α -Te nanosheets and β -Te nanoribbons. a-g Sample thickness distribution statistics for α -Te nanosheets and β -Te nanoribbons under different gas flows: (a) 50 sccm, (b) 80 sccm, (c) 100 sccm, (d) 125 sccm, (e) 150 sccm, (f) 175 sccm, and (g) 200 sccm. **h** Thickness of α -Te nanosheets and β -Te nanoribbons as a function of the Ar flow rate.

2 On Pg9 paragraph 2: “consistent with PDF#97-005-2499 and PDF#36-1452” What does “PDF#xxx” mean?

RESPONSE and CHANGES: We thank the reviewers for their careful review of our manuscript. PDF#97-005-2499 in the manuscript refers to the α -Te phase structure derived from Jade's calculations from ICSD Serial No. 52499, where "97" refers to the International Centre for Diffraction Data (ICDD) data source and "005" and "2499" correspond to the volume and number of the collection, respectively. Similarly, PDF#36-1452 refers to the β -Te phase structure derived from the International Center for Diffraction Data (ICDD) data source in card number 1452 of volume 36.

3. The unit cell of beta Te in Fig. 1 and Supplementary 1b is inconsistent with the structure reported in other literatures and the symmetry group. Beta Te has a 1D van der Waals structure with three-fold screw symmetry. Each tellurium atom forms 2 covalent bonds with 2 neighboring Te atoms within the chain (e. g. Nano Lett. 2017, 17, 6, 3965). However, in the manuscript, there is an additional interchain bonding, which makes three Te atoms inequivalent, and the crystal no longer belongs to $P3_121$ symmetry group.

RESPONSE: We thank the reviewer for your careful reading of our manuscript and for the discussion on the structure of the material. Indeed, its bulk phase structure is consistent with that in previous literature (*Nano Lett.* 2017, **17**, 6, 3965), maintaining the $P3_121$ symmetry group. Here, the monolayer structure is used for the calculations and illustrations in our work to simulate the interfacial interactions between β -Te and the substrate under experimental conditions. The structural arrangement of the β -Te monolayer used in our calculations is consistent with that in previous literature (*Phys. Rev. Lett.* 2017, **119**, 106101; *Nature* 2017, **552**, 40-41; *Phys. Rev. B* 2019, **99**, 195436; *Appl. Phys. Lett.* 2019, **115**, 151104).

4. Pg 14, the calculated bulk bandgap of beta Te is 0.09 eV, which is significantly lower than that reported in the literature (0.3-0.35 eV; e.g., *ACS Nano* 2018, 12, 7, 7253). Please clarify.

RESPONSE: We thank the reviewer for noting this important point. In our manuscript, the bandgap was determined based on calculations using the Perdew-Burke-Ernzerhof (PBE) functional, and the bulk bandgap of β -Te was found to be 0.09 eV. Our calculated bandgap results are similar to those in the literature, which also employ the PBE functional or GGA-SCAN functional (*Phys. Rev. B* 2007, **75**, 245437; *Inorg. Chem.* 2018, **57**, 5083-5088). Moreover, we performed additional calculations using the Heyd-Scuseria-Ernzerhof functional with the spin-orbit coupling method (HSE + SOC), and as shown in Fig. R2a, the bulk bandgap of β -Te obtained with HSE + SOC was approximately 0.34 eV, in good agreement with the experimental results in previous literature (*ACS Nano* 2018, **12**, 7253-7263). As shown in Fig. R2, we also performed HSE + SOC functional calculations on monolayer, trilayer and bulk β -Te for bandgap determination. As the number of layers decreased, the bandgap of β -Te gradually increased, which was consistent with the results of the PBE calculations. An increase in the number of atoms in multilayer structures leads to a substantial escalation in computational demands. Due to the intricacies of electronic interactions, this escalation is not linear. Consequently, for computationally intensive HSE + SOC band structure calculations, the vast majority of computational resources are unable to support the

computation of structures with more layers. Considering the burden of calculation and the similarity of band structures between PBE and HSE + SOC, we used only the PBE functional in our manuscript to investigate the influence of material thickness on the band structure of Te. We have added corresponding explanations in the revised manuscript (highlighted in red, page 14, line 292, Supplementary Fig. 12).

Fig. R2 Calculated band structure and bandgap of β -Te. a Calculated band structure of bulk β -Te.

The purple lines are calculated using the PBE functional, and the blue lines are calculated using the HSE functional with the SOC method. The inset is the Brillouin zone. **b** Bandgaps of β -Te nanoribbons as a function of the number of atomic layers using HSE+SOC functional calculations.

5. Pg 15: The authors extracted an electron mobility of $74.7 \text{ cm}^2/\text{Vs}$ in alpha Te. However, there is only a small on/off ratio with the gate. While the semiconducting or semi-metallic behavior in thin alpha Te is somewhat interesting, it should be noted that it is not a technically a transistor and thus using the terminologies in FETs seems out of place. The field-effect mobility should be extracted in a region where dI_{ds}/dV_{gs} is constant over a sufficiently large window of V_{gs} . In addition, a low V_{ds} should be used to ensure that the device is in the linear regime.

RESPONSE and CHANGES: Thank you for the reminders and valuable suggestions. We have modified the description of “the α -Te FET” to “the α -Te device” in the revised manuscript. Moreover, as V_g varies from -60 V to 60 V, the dI_{ds}/dV_{gs} of the α -Te device reaches a relatively steady state (Fig. R3a). We obtained a maximum mobility of $74.7 \text{ cm}^2\text{V}^{-1}\text{s}^{-1}$ at -60 V in our manuscript ((Fig. R3b), which is consistent with the

calculation methods for the mobility of previously reported semimetallic two-dimensional materials (*ACS Nano* 2018, **12**, 4055-4061; *Nat. Nanotech.* 2010, **4**, 487-496).

Fig. R3 **Electrical characterization of α -Te.** **a** dI_{ds}/dV_{gs} for the α -Te device at $V_g = -60$ to 60 V. **b** The mobility of the α -Te device is plotted against V_g .

In addition, we tested the I_{ds} - V_{gs} curves of another α -Te device at different V_{ds} . The linear regime of the device at low V_{ds} is consistent with the test results at high V_{ds} , as shown in Fig. R4. The mobility (μ) changes are not significant at either high or low V_{ds} ($V_{ds} = 0.2$ V, $\mu = 38.94$ $\text{cm}^2 \text{V}^{-1} \text{s}^{-1}$ vs $V_{ds} = 2$ V, $\mu = 36.19$ $\text{cm}^2 \text{V}^{-1} \text{s}^{-1}$), as shown in Table R1.

Fig. R4 **Transfer curves of the α -Te device at different bias voltages.** **a** Transfer curves of the 1.9 nm thick α -Te device at $V_{ds} = 0.5$ - 2 V. **b** Transfer curves of the 1.9 nm thick α -Te device at $V_{ds} = 0.2$ - 0.5 V.

Table R1 Mobility at different V_{ds} for the same α -Te device.

dI_{ds}/dV_{gs}	L (μm)	W (μm)	V_{ds} (V)	C_g (F)	μ ($\text{cm}^2 \text{V}^{-1} \text{s}^{-1}$)
9.39×10^{-8}	2	2.095	0.2	1.15×10^{-8}	38.97

1.29×10^{-7}	2	2.095	0.3	1.15×10^{-8}	35.70
1.67×10^{-7}	2	2.095	0.4	1.15×10^{-8}	34.66
1.96×10^{-7}	2	2.095	0.5	1.15×10^{-8}	32.54
3.97×10^{-7}	2	2.095	1	1.15×10^{-8}	32.96
6.02×10^{-7}	2	2.095	1.5	1.15×10^{-8}	33.32
8.72×10^{-7}	2	2.095	2	1.15×10^{-8}	36.19

6. *Multiple device electrical characteristics were displayed in the manuscript and Supplementary. I recommend the authors to add key device parameters such as thickness, W , and L to each figure, or number the devices so that the readers can easily follow.*

RESPONSE and CHANGES: Thank you for your patience in reviewing this manuscript and for your very valuable suggestions. In the revised manuscript, we have labeled all the devices sequentially and given important parameters such as thickness, W , and L of the devices in detail, as shown in Table R2.

We added the corresponding figure and table to Supplementary Fig. 14 and Extended Data Table 1, respectively.

Fig. R5 Electrical characterization of β -Te transistors.

Table R2 The key device parameters for β -Te transistors

Sample	Substrate	V_{ds} (V)	L (μm)	W (nm)	Thickness (nm)	Current density (μA μm^{-1})
B1	SiO ₂	-2	0.92	16.2	1.5	985.02
B2	SiO ₂	-3	0.92	27	1.9	1050.89
B3	SiO ₂	-1.5	0.98	33	3.5	1018
B4	SiO ₂	-3	0.78	77	2.1	1199.74

B5	SiO ₂	-4	0.90	47	3.3	1043.83
B6	SiO ₂	-3	0.99	49.6	3.7	1042.81
B7	h-BN/Si ₃ N ₄	-6	1.22	29.3	5.2	1276.4
B8	h-BN/Si ₃ N ₄	-4	1.02	36	14.3	1527
B9	h-BN/Si ₃ N ₄	-1.5	0.066	44	10	1325
B10	h-BN/Si ₃ N ₄	-1/-1.5	0.046	35.6	8.0	1270/1460

7. Page 15: The current density in beta Te is certainly impressive. However, a potential pitfall is the way the authors normalize the current with width. Can the authors provide the information of the channel width W for these devices and how they are measured? Assuming a cylindric morphology of these nanoribbons, the diameter would be only a few nm, which is hard to accurately measure. More importantly, in Eq (4) the authors used a model to calculate the capacitors where an effective channel width is defined by a hyperbolic equation, and it can lead to over-estimation of current density and mobility.

RESPONSE: Thank you for your suggestions. We divided the device's current by the channel width (W); this extraction method is valid and consistent with previous literature reports (*Nat. Electron.* 2018, **1**, 228-236; *Nature* 2023, **613**, 274-279; *Nature* 2023, **616**, 470-475). We used scanning electron microscopy (SEM) to accurately measure the values of the electrical parameters (L , W) of each β -Te nanoribbon device, and to ensure the accuracy of W , we performed at least five measurements and obtained the average values, as shown in Fig. R6. The detailed electrical performance-related parameters of all the β -Te nanoribbon FETs in our work were accurately measured and quantified, as shown in Table R2.

Fig. R6 Characterization of the channel width W and length L of β -Te nanoribbon FETs.

In addition, our β -Te nanoribbons do not exhibit a cylindrical morphology. Most of the width of β -Te ranges from 22 nm to 94 nm (Fig. R7a-b), which was accurately measured using scanning electron microscopy (SEM). The thickness of β -Te ranges from 0.9 nm to 8.9 nm and was precisely measured using atomic force microscopy (AFM), as shown in Fig. R7c. We performed detailed statistics on the width and thickness distributions. The width and thickness distributions were determined, as shown in Fig. R7d-e. The experiment confirmed that the synthesized β -Te was in nanoribbon form rather than cylindrical nanowires, and Equation 4 was used for calculating its mobility. Using the formula (Eq. 1) for β -Te nanoribbons reported in the literature to calculate the mobility of β -Te nanoribbons in our article (*Nat. Electron.* 2018, **1**, 228-236; *Adv Mater.* 2018, **30**, 1803109; *Mater. Today*, 2023, **63**, 50-58). The maximum value reaches $3752.27 \text{ cm}^2 \text{ V}^{-1} \text{ s}^{-1}$, which is significantly greater than that reported in the referenced articles. However, considering that our device's channel width is smaller than the thickness of the dielectric layer, to ensure the accuracy of the mobility, this paper employs the nanowire mobility calculation formula (Eq 4) for estimation, as shown in Table R3.

Fig. R7 Micromorphological characterization of β -Te nanoribbons. a Scanning electron morphology. **b** Electron microscopy images. **c** Atomic force morphology. **d, e** Relative frequency distribution diagram of width (**d**) and thickness (**e**).

8. Following Q7, Eq. (4) is typically used to calculate the parasitic capacitance of a cylindrical wire over a ground plane in a uniform dielectric. I am skeptical whether this is applicable to this device geometry since the nanowire is in air whereas the gate is isolated by SiO₂. A more rigid validation of capacitance calculation needs to be provided since it is critical to back up the key claim of superior electrical performance of p-type beta Te.

RESPONSE: Thank you for considering the depth of our electrical performance of p-type beta Te. Eq. (4) also applies to narrow nanoribbons (*Nano Lett.* 2007, **7**, 2463-2469; *Adv. Mater.* 2003, **15**, 143-146; *J. Phys. Chem. B* 2000, **104**, 5213-5216.). If we use the formula (Eq. 1) for β -Te nanoribbons reported in the literature (the capacitance originates from SiO₂; *Nat. Electron.* 2018, **1**, 228-236; *Adv Mater.* 2018, **30**, 1803109; *Mater. Today* 2023, **63**, 50-58) to calculate the mobility of β -Te nanoribbons in our article. The maximum value reaches 3752.27 cm² V⁻¹ s⁻¹, which is significantly greater than that reported in the referenced articles. However, considering that our device's channel width is smaller than the thickness of the dielectric layer, to ensure the accuracy of the mobility, we adopted a more conservative approach for mobility calculation (Eq.

4) with a maximum value of $690.69 \text{ cm}^2 \text{ V}^{-1} \text{ s}^{-1}$.

Table R3 Mobility of β -Te devices under different computational formulas.

Number	L (μm)	W (μm)	$ V_{ds} $	dI_{ds}/dV_g	μ of Eq. 1 μ ($\text{cm}^2 \text{ V}^{-1} \text{ s}^{-1}$)	μ of Eq. 4 ($\text{cm}^2 \text{ V}^{-1} \text{ s}^{-1}$)
1	5.181	0.07621	1	2.03×10^{-7}	1198.27	218.57
2	5.522	0.1414	1	2.78×10^{-7}	942.35	278.98
3	5.522	0.1414	1.5	4.04×10^{-7}	914.16	211.87
4	4.839	0.058	0.5	6.15×10^{-8}	886.25	149.44
5	2.961	0.0468	2	3.93×10^{-7}	1078.55	136.85
6	3.949	0.086	1	1.75×10^{-7}	698.67	135.23
7	2.88	0.0192	0.1	1.75×10^{-8}	2286.52	149.26
8	3.75	0.0513	2	5.2×10^{-7}	1652.05	223.45
10	3.064	0.0189	0.4	6.37×10^{-8}	2235.97	144.92
11	3.979	0.0886	0.5	1.2×10^{-7}	936.82	185.09
12	3.198	0.06168	0.5	1.14×10^{-7}	1023.43	158.27
13	2.704	0.02155	0.4	1.01×10^{-7}	2760.46	196.73
14	4.729	0.0829	0.7	1.55×10^{-7}	1098.37	207.38
15	6.152	0.1316	0.4	1.27×10^{-7}	1287.59	331.32
16	4.853	0.1108	0.1	1.96×10^{-8}	745.35	171.21
17	6.09	0.0864	0.1	1.3×10^{-8}	794.96	154.72
18	6.224	0.0978	0.7	2.23×10^{-7}	1758.99	176.17
19	5.968	0.2128	0.1	3.17×10^{-8}	773.06	269.03
20	6.228	0.111	1.5	6.32×10^{-7}	2055.67	350.54
21	5.898	0.1972	2.5	2.0×10^{-6}	2080.60	690.69
22	5.969	0.0956	0.5	2.12×10^{-7}	2306.37	254.27
23	5.889	0.09618	3	1.65×10^{-6}	2924.79	621.89
24	5.972	0.1082	3	7.01×10^{-7}	1121.47	185.92
25	5.397	0.0877	2.5	7.75×10^{-7}	1658.88	211.19

26	4.361	0.0690	1	1.29×10^{-7}	707.25	117.98
27	3.009	0.149	2	4.61×10^{-7}	405.12	112.57
28	0.9182	0.01629	0.1	4.78×10^{-8}	2342.86	135.02
29	0.9278	0.01288	1	4.09×10^{-7}	2561.91	122.99
30	0.9	0.047	0.4	4.33×10^{-7}	1802.5	228.94
31	0.92	0.027	0.4	2.52×10^{-7}	1866.67	157.98
32	0.9931	0.04962	0.7	5.40×10^{-7}	1342.56	177.56
33	0.975	0.03031	1	6.32×10^{-7}	1767.82	163.07
34	0.9193	0.02477	1.5	9.57×10^{-7}	2058.99	163.34
35	1.022	0.036	1	1.52×10^{-6}	3752.27	567.80

9. WS₂ should be semiconducting (see *Nat. Commun.* 12, 693, 2021), while in Supplementary Fig. 14, the WS₂ template is completely insulating. Why is that?

RESPONSE and CHANGES: Thanks to the reviewers' thorough consideration of our work, we have added additional data to confirm the electrical properties of monolayer WS₂. As shown in Fig. R8a, The PL spectrum indicates that monolayer WS₂ is a typical semiconductor material. However, our monolayer WS₂ samples exhibit undesirable electrical properties with currents mainly in the range of 6×10^{-10} to 4×10^{-13} A (Fig. R8b-f), which is due to the presence of too many defects in the monolayer WS₂ substrate obtained by NaCl-assisted chemical vapor deposition (CVD) and secondary high-temperature treatment as the growth substrate. The current values ($10^{-6} \sim 10^{-4}$ A) of the Te samples we collected are greater than those of WS₂. The insulating WS₂ possesses an atomically flat surface, providing an ideal platform for the growth of high-quality single-crystal Te samples. Due to the absence of dangling bonds on the WS₂ nanosheet surface, which results in a low charge scattering center density and weaker charge trapping states, β -Te exhibits superior electrical properties.

We added the corresponding information to Supplementary Fig. 16 (highlighted in red, page 49, line 830).

Fig. R8 Characterization of the optical and electrical properties of monolayer WS₂. a

Photoluminescence (PL) spectrum of monolayer WS₂. **b, c** Output (**b**) and transfer (**c, d**) curves ($V_{ds} = -0.1, 1$ V) of Sample 1. **e, f** Transfer curves ($V_{ds} = -0.5$ V) of Sample 2 and Sample 3, respectively.

10. Pg. 16 the authors inferred the on-state current increase after 5 months is due to measurement error. Can the authors elaborate what measurement errors?

RESPONSE and CHANGES: We thank the reviewers for their critical review. The device current density slightly increased after 5 months (9.9×10^{-4} A for 0 months vs 1.1×10^{-3} A for 5 months). Its numerical variation is not significant at $V_{ds} = -2$ V due to the difference in the measurement frequency. The corresponding modifications were also made in the revised manuscript on page 17, line 340 (highlighted in red).

Fig. R9 Comparison of transmission curves ($V_{ds} = -2$ V) for β -Te FETs at 0 days and 5 months.

11. Pg. 18 *Is the field effect mobility extracted from short-channel devices? It should be noted that in short-channel devices with a universal back gate, the field-effect mobility is often over-estimated because the gate is also modulating contacts. The field-effect mobility should only be extracted in long channel devices where the channel resistance is significantly larger than the contact resistance. The authors should report mobility with long channel devices where channel resistance is sufficiently larger than contact resistance, or may just report transconductance instead.*

RESPONSE and CHANGES: We thank the reviewers for their insightful questions on field effect mobility. In fact, the mobility of our β -Te FETs is determined from the long channel (0.9-5.5 μm), and the detailed calculated parameters of mobility for some of the devices are shown in Table R3.

Reviewer #2

This manuscript demonstrated phase engineering of tellurium and also showed some high performance devices made out of the Te. The authors realized different phases of Te by controlling the flow rate, which was not reported before, and showed systematically evolutionary study of this process. However, the importance of this work is not clearly manifested, the structure of this manuscript is not well organized, and is not of scientific rigorous. I think this manuscript should at least have a major revise or try other specific journals which focus on growth.

RESPONSE: We are very grateful to the reviewers for taking their valuable time to review our manuscript and for their positive comments, such as "The authors realized different phases of Te by controlling the flow rate, which was not reported before, and showed systematic evolutionary study of this process". We have added experimental data from this work and restructured the manuscript to highlight the significance of the work. Below, we respond to each of the comments and indicate specific changes to the manuscript and additional information.

1. Although the manuscript showed realization of the phase engineering of Te, it didn't demonstrate the application of the different phases, especially the application of phase transition. So, the question raises -- what is the significance of realizing different phases (refer to MoTe2)?

RESPONSE and CHANGES: Thank you for your suggestion. We have added a description of the significance of phase engineering in the revised manuscript.

Tellurium (Te) has recently attracted much attention in optoelectronic devices due to its unique helical chain structure, unusual anisotropic crystal structure, tunable bandgap and unusually low thermal conductivity. Two-dimensional (2D) Te may have different crystal structures with a variety of bonding configurations. Among them, the α -Te phases exhibit a mixture of triplet and sixfold coordination, while β -Te exhibits a mixture of triplet and quadruple coordination, which have very different physical and electronic properties. The α -Te configuration corresponds structurally to the 1T configuration commonly adopted for 2D materials known as transition metal dichalcogenides (TMDs), which possess inversion symmetry and semimetallic electronic structures with complex topological states. Moreover, α -Te exhibits a layer-dependent bandgap, low exciton binding energy (~ 0.18 eV), extraordinary mobility, high optical absorption, and strong infrared oscillation intensity, which holds broad prospects for applications in sensitive photodetectors. β -Te is a p-type semiconductor with a bandgap of ~ 0.3 eV, and due to spin-orbit coupling, it possesses a small effective mass and a high hole mobility of up to several thousand $\text{cm}^2 \text{V}^{-1} \text{s}^{-1}$. Meantime, β -Te possesses rich and intriguing characteristics, such as photoconductivity, thermoelectricity, and piezoelectricity, which demonstrates significant potential applications in various fields, including photodetectors, field-effect transistors, piezoelectric devices, modulators, and energy harvesting devices. The dynamic control of α -Te and β -Te phase engineering can reveal the competition, coexistence, and cooperation among different crystal structures as well as the interactions between different physical properties. Therefore, phase engineering between semimetallic α -Te and semiconducting β -Te polymorphs holds significant importance for widespread

device applications, such as phase-engineering memory devices, high-performance transistors, reconfigurable circuits, and topological transistors, under atomically thin limits.

To date, p-type β -Te materials with excellent electrical properties have been successively synthesized using various methods, such as hydrothermal synthesis, molecular beam epitaxy (MBE), thermal evaporation, atomic layer deposition (ALD), and chemical vapor deposition (CVD). Although Wu et al. synthesized β -Te nanosheets with mobilities of approximately $700 \text{ cm}^2 \text{ V}^{-1} \text{ s}^{-1}$ and a significant ON-state current density of $1.06 \text{ mA } \mu\text{m}^{-1}$ at $V_{\text{ds}} = -1.4 \text{ V}$ using a substrate-free solution, their ON/OFF ratio was only 10^1 . In addition, Zhao et al. reported the fabrication of β -Te thin films by thermal evaporation at room temperature with an ON/OFF current ratio of $\sim 10^4$, but an average mobility was $21.1 \text{ cm}^2 \text{ V}^{-1} \text{ s}^{-1}$, an ON-state current was approximately $0.001 \text{ mA } \mu\text{m}^{-1}$. However, these previous reports have been limited to the synthesis of β -Te, and phase engineering between semimetallic α -Te and semiconducting β -Te has not been reported to date, especially at the monolayer limit. Exploring phase transition competition at the atomic scale is crucial because it involves significantly enhanced electron-phonon and electron-electron interactions.

Here, we designed an atomic cluster density- and interface-guided multiple controllable strategy for the phase- and thickness-controlled synthesis of monolayer α -Te nanosheets and β -Te nanoribbons. The α -Te nanosheets exhibit a transition from a metal to an n-type semiconductor, whereas the β -Te nanoribbons are always p-type semiconductors as the thickness decreases, and both of them have good air stability after several months. Furthermore, the reduced thickness restricts the transport of carriers in the atomic channel and causes the β -Te nanoribbon to exhibit an exceptional ON-state current density of $\sim 1527 \text{ } \mu\text{A } \mu\text{m}^{-1}$ and a mobility as high as $\sim 690.7 \text{ cm}^2 \text{ V}^{-1} \text{ s}^{-1}$ at room temperature, which suggests that 2D Te could be applied in electronic devices that offer higher performance than those that use MoS_2 .

By employing the continuous vdW epitaxy growth of different phases of Te, achieving heteroepitaxial integration of metal-semiconductor states within the same atomic plane has become feasible, enabling atomic-scale metal-semiconductor contacts

and reducing contact resistance. Therefore, the realization of Te phase engineering has paved the way for enhancing the performance of optoelectronic devices, wearable electronics, synaptic devices, and friction-based nanogenerators, laying the groundwork for large-scale, high-performance integrated circuits.

The significance of realizing different phases of Te and the application of phase transition have been added to the Introduction (highlighted in red, page 2, line 28) and Discussion (highlighted in red, page 20, line 407) of the revised manuscript.

*2. Te is of particularly interesting because it could be used for p-type high performance transistors, especially for back-end-of-line integration. The devices in this manuscript did not achieve particularly superior performance for this purpose. The mobility was not as high as previously reported ($>700 \text{ cm}^2/\text{V}\cdot\text{s}$, *Nature Electronics* 1, 228(2018), *Advanced Materials* 30, 1803109 (2018), *Mater. Today* 63, 50-58 (2023)), and it did not achieve a particularly superior improvement in leakage current nor a fully back-end-of-line compatible preparation temperature, as demonstrated in paper (*Applied Surface Science* 636, 157801 (2023))*

RESPONSE and CHANGES: Thank you for your insightful questions. The article's innovation lies in the following aspects:

(1) If we use the formula reported in the literature to calculate the mobility of β -Te nanoribbons (*Nat. Electron.* 2018, **1**, 228-236; *Adv Mater.*2018, **30**, 1803109; *Mater. Today* 2023, **63**, 50-58). The maximum value reaches $3752.27 \text{ cm}^2 \text{ V}^{-1} \text{ s}^{-1}$; in Table R3, this value is significantly greater than the reported mobility of β -Te ($1755 \text{ cm}^2 \text{ V}^{-1} \text{ s}^{-1}$ in *Mater. Today* 2023, **63**, 50-58). As the β -Te width is smaller than the dielectric layer thickness, to ensure the accuracy of the electrical performance and avoid overstating device capabilities, we utilized the more conservative calculation formula in this manuscript to calculate a mobility of $\sim 690.7 \text{ cm}^2 \text{ V}^{-1} \text{ s}^{-1}$, which is still comparable to the reported mobility ($\sim 700 \text{ cm}^2 \text{ V}^{-1} \text{ s}^{-1}$ in *Nat. Electron.* 2018, **1**, 228-236; *Adv Mater.*2018, **30**, 1803109). Additionally, according to the International Roadmap for Devices and Systems (IRDS), for semiconductor metal-oxide-semiconductor (MOS) field-effect transistors (FETs), the final power supply voltage will be $\leq 1.0 \text{ V}$. At $V_{\text{ds}} =$

-1 V, the current density of the β -Te FET in our article is $1.27 \text{ mA } \mu\text{m}^{-1}$, which is significantly higher than the reported β -Te current density ($\sim 0.85 \text{ mA } \mu\text{m}^{-1}$ at $V_{\text{ds}} = -1 \text{ V}$).

(2) Furthermore, because the synthesized β -Te particles are relatively thick, their ON/OFF ratio is very small ($\sim 10^1$ - 10^2 in *Adv Mater.* 2018, **30**, 1803109; *Mater. Today* 2023, **63**, 50-58). Although Wu et al. reported a current density of $1.06 \text{ mA } \mu\text{m}^{-1}$ for 11.1 nm thick β -Te at $V_{\text{ds}} = -1.4 \text{ V}$, their ON/OFF ratio was only 10^1 (*Nat. Electron.* 2018, **1**, 228-236). In contrast, the majority of our β -Te devices maintain ON/OFF ratios of 10^3 - 10^4 . Particularly in narrow-channel devices (46 nm), at $V_{\text{ds}} = -1.5 \text{ V}$, the current density reaches $\sim 1.5 \text{ mA } \mu\text{m}^{-1}$, while the ON/OFF ratio still remains at 10^3 .

(3) Although Park et al. synthesized Te thin films using thermal evaporation at room temperature (*Appl. Surf. Sci.* 2023, **636**, 157801), these films were amorphous, limiting their electrical performance (average mobility of $21.1 \text{ cm}^2\text{V}^{-1}\text{s}^{-1}$ and current density of $\sim 0.001 \text{ mA } \mu\text{m}^{-1}$ at $V_{\text{ds}} = -8 \text{ V}$). In our manuscript, although the growth temperature of Te is relatively high (310-550 °C), its overall electrical performance is excellent (hole mobility as high as $\sim 690.7 \text{ cm}^2 \text{ V}^{-1} \text{ s}^{-1}$, a high I_{ON} of $1.27 \text{ mA } \mu\text{m}^{-1}$ at $V_{\text{ds}} = -1 \text{ V}$, and excellent electrical air stability maintained after 5 months). Furthermore, the endeavor to synthesize high-performance, low-temperature p-type Te for back-end-of-line integration has been our ongoing pursuit.

(4) We can grow large-area, highly oriented Te single crystals. The highly single-orientation (90%) epitaxy of β -Te nanoribbons on monolayer 2-inch MoS_2 provides a reliable path for the future preparation of large-area, high-quality, and scalable 2D Te arrays and single-crystal films. This has laid a solid foundation for the future integration and industrial application of 2D Te devices.

(5) Most of the existing 2D materials suffer from either low carrier mobility (such as MoS_2 and MoTe_2) or poor air stability (such as BP). Therefore, identifying a 2D material that combines the advantages of high carrier mobility, good air stability, and easy fabrication is vital to the development of electronic devices in the next generation. Our β -Te nanoribbons exhibit excellent air-stable p-type electrical performance, while α -Te nanosheets demonstrate outstanding air-stable semimetal properties, which

enables the possibility of achieving metal-semiconductor heteroepitaxial integration within the same atomic plane by leveraging phase engineering of Te, thereby achieving atomic-scale metal-semiconductor contacts and reducing contact resistance. As a result, the realization of Te phase engineering lays the foundation for enhancing optoelectronic devices and facilitating large-scale, high-performance back-end-of-line integrated circuits.

Table R3 Mobility of β -Te devices under different computational formulas.

Number	L (μm)	W (μm)	$ V_{\text{ds}} $	Dids/Dvg	Updated μ ($\text{cm}^2 \text{V}^{-1} \text{s}^{-1}$)	μ of Eq. 4 ($\text{cm}^2 \text{V}^{-1} \text{s}^{-1}$)
1	5.181	0.07621	1	2.03×10^{-7}	1198.27	218.57
2	5.522	0.1414	1	2.78×10^{-7}	942.35	278.98
3	5.522	0.1414	1.5	4.04×10^{-7}	914.16	211.87
4	4.839	0.058	0.5	6.15×10^{-8}	886.25	149.44
5	2.961	0.0468	2	3.93×10^{-7}	1078.55	136.85
6	3.949	0.086	1	1.75×10^{-7}	698.67	135.23
7	2.88	0.0192	0.1	1.75×10^{-8}	2286.52	149.26
8	3.75	0.0513	2	5.2×10^{-7}	1652.05	223.45
10	3.064	0.0189	0.4	6.37×10^{-8}	2235.97	144.92
11	3.979	0.0886	0.5	1.2×10^{-7}	936.82	185.09
12	3.198	0.06168	0.5	1.14×10^{-7}	1023.43	158.27
13	2.704	0.02155	0.4	1.01×10^{-7}	2760.46	196.73
14	4.729	0.0829	0.7	1.55×10^{-7}	1098.37	207.38
15	6.152	0.1316	0.4	1.27×10^{-7}	1287.59	331.32
16	4.853	0.1108	0.1	1.96×10^{-8}	745.35	171.21
17	6.09	0.0864	0.1	1.3×10^{-8}	794.96	154.72
18	6.224	0.0978	0.7	2.23×10^{-7}	1758.99	176.17
19	5.968	0.2128	0.1	3.17×10^{-8}	773.06	269.03
20	6.228	0.111	1.5	6.32×10^{-7}	2055.67	350.54

21	5.898	0.1972	2.5	2.0×10^{-6}	2080.60	690.69
22	5.969	0.0956	0.5	2.12×10^{-7}	2306.37	254.27
23	5.889	0.09618	3	1.65×10^{-6}	2924.79	621.89
24	5.972	0.1082	3	7.01×10^{-7}	1121.47	185.92
25	5.397	0.0877	2.5	7.75×10^{-7}	1658.88	211.19
26	4.361	0.0690	1	1.29×10^{-7}	707.25	117.98
27	3.009	0.149	2	4.61×10^{-7}	405.12	112.57
28	0.9182	0.01629	0.1	4.78×10^{-8}	2342.86	135.02
29	0.9278	0.01288	1	4.09×10^{-7}	2561.91	122.99
30	0.9	0.047	0.4	4.33×10^{-7}	1802.5	228.94
31	0.92	0.027	0.4	2.52×10^{-7}	1866.67	157.98
32	0.9931	0.04962	0.7	5.40×10^{-7}	1342.56	177.56
33	0.975	0.03031	1	6.32×10^{-7}	1767.82	163.07
34	0.9193	0.02477	1.5	9.57×10^{-7}	2058.99	163.34
35	1.022	0.036	1	1.52×10^{-6}	3752.27	567.80

3. The author also did not introduce some of the most recent and important Te research results in the introduction. There have been many reports on the high-quality growth of Te using different methods and realization of different phases and the study of devices, not limited to the reference mentioned above.

RESPONSE and CHANGES: Thank you for the detailed suggestions. We have incorporated the information regarding Te using different methods and realization of different phases and the study of devices into the introduction of revised manuscript (highlighted in red, page 2, line 28).

As a new type of elemental two-dimensional (2D) material, monolayer Te (referred to as 'tellurene') was introduced in 2017 and has garnered increasing attention in recent years. Compared to graphene, Te possesses a bandgap of ~ 0.3 eV. It exhibits excellent air stability compared to phosphorene and demonstrates higher carrier mobility (several thousand $\text{cm}^2 \text{V}^{-1} \text{s}^{-1}$) as well as a thinner single-layer thickness (~ 0.35 nm) compared

to two-dimensional transition metal dichalcogenides. These characteristics make it suitable for meeting the demands of high integration, low power consumption, and integration into ultrashort-channel devices. For example, Zhang et al. first predicted several stable tellurene allotropes at the few-layer or monolayer limit, including 1T-MoS₂-like (α -Te), metastable tetragonal (β -Te), and 2H-MoS₂-like (γ -Te) structures, by ab initio simulations; subsequently, successful synthesis of β -Te on a graphite substrate was achieved. (*Phys. Rev. Lett.* 2017, **119**, 106101; *Nanoscale* 2017, **9**, 15945-15948). Subsequently, p-type β -Te, which has excellent electrical properties, was successively synthesized using various methods, such as hydrothermal synthesis (*Nano Lett.* 2017, **17**, 3965-3973; *Nat. Electron.* 2018, **1**, 228-236), molecular beam epitaxy (MBE) (*Nano Lett.* 2017, **17**, 4619-4623; *Adv Mater.* 2018, **30**, 1803109), thermal evaporation (*Nat. Nanotech.* 2020, **15**, 53-58; *Appl. Surf. Sci.* 2023, **636**, 157801), atomic layer deposition (ALD) (*ACS Nano* 2023, **17**, 15776-15786), and chemical vapor deposition (CVD) (*Nano-Micro Lett.* 2022, **14**:109; *Mater. Today* 2023, **63**, 50-58). Although Wu et al. synthesized β -Te nanosheets with mobilities of approximately 700 cm² V⁻¹ s⁻¹ and a significant ON-state current density of 1.06 mA μ m⁻¹ at V_{ds} = -1.4 V using a substrate-free solution, their ON/OFF ratio was only 10¹ (*Nat. Electron.* 2018, **1**, 228-236). In addition, Zhao et al. reported the fabrication of β -Te thin films by thermal evaporation at room temperature with an ON/OFF current ratio of $\sim 10^4$, but an average mobility was 21.1 cm² V⁻¹ s⁻¹, an ON-state current was approximately 0.001 mA μ m⁻¹. (*Appl. Surf. Sci.* 2023, **636**, 157801). However, these previous reports have been limited to the synthesis of β -Te, and phase engineering between semimetallic α -Te and semiconducting β -Te has not been reported to date, especially at the monolayer limit.

Here, we designed an atomic cluster density- and interface-guided multiple controllable strategy for the phase- and thickness-controlled synthesis of monolayer α -Te nanosheets and β -Te nanoribbons. The gas flow rate can impact the formation of atomic clusters and substrates, and the use of monolayer WS₂ provides favorable growth conditions for the adsorption and surface diffusion of Te atoms, thereby regulating the synthesis of the Te phase. In addition, we synthesized p-type semiconductor β -Te nanoribbons with outstanding comprehensive electrical properties

(hole mobility as high as $\sim 690.7 \text{ cm}^2 \text{ V}^{-1} \text{ s}^{-1}$, high I_{ON} of $\sim 1527 \mu\text{A} \mu\text{m}^{-1}$ ON/OFF ratio of 10^4) and excellent air stability after 5 months.

4. On page 3, paragraph 2, the authors wrote that vdW epitaxial growth is a newly developed technology, but in fact this technology has been reported at least in the 1990s. In addition, the authors mentioned vdW epitaxy many times in the manuscript, however, based on the growth structures, the growth method employed should belong to the concept of remote epitaxy.

RESPONSE and CHANGES: Thank you for your insightful opinion. van der Waals (vdW) epitaxial growth was indeed reported in the 1990s, mainly focusing on traditional 3D materials. In recent years, with the vigorous development of two-dimensional materials, vdW epitaxy has gradually been applied to the synthesis of 2D materials. Therefore, we mentioned that vdW epitaxial growth has recently been used to specifically target emerging 2D materials. We consider that the growth of Te on the WS_2 substrate results in vdW epitaxy for the following reasons: vdW epitaxial growth relies on van der Waals forces to control the growth of thin films, which are weak intermolecular forces that arise from interactions between molecules in 2D materials. In contrast, remote epitaxy utilizes the partially screened potential between the substrate beneath the 2D material and the growing atoms to achieve indirect interactions. This remote interaction can result in the growth of epitaxial films that are oriented with respect to the substrate, without the need for direct chemical bonding between the 2D material and the substrate. One of the most commonly used 2D materials for remote epitaxy is graphene, which is used for growing three-dimensional film materials such as GaAs and GaN (*Nat. Electron.* 2019, **2**, 439-450.; *Nature* 2017, **544**, 340-343.; *Sci. Adv.* 2023, **9**, eadj5379). The interaction between Te and the WS_2 substrate is governed by van der Waals forces

Therefore, we consider the growth of Te on a WS_2 substrate to be vdW epitaxial growth. In our manuscript, phase engineering of atomically thin α -Te nanosheets and β -Te nanoribbons with high purity and crystallinity is realized by a vdW epitaxy method. The gas flow rate can impact the formation of atomic clusters and substrates, and the

use of monolayer WS₂ provides favorable growth conditions for the adsorption and surface diffusion of Te atoms, thereby regulating the synthesis of the Te phase. At low flow rates (50-125 sccm), the lower atomic cluster density caused by the long distance between Te atoms leads to fewer interactions among these atoms. The formation of α -Te atom clusters with a substrate (WS₂) can be decreased, facilitating the growth of α -Te at this time. At high flow rates (125-200 sccm), the cluster density increases, facilitating the formation of the β -Te phase.

[REDACTED]

Fig. R10 Schematic representation of different epitaxial growth processes (*Nat. Electron.*2019, **2**, 439-450.). **a** van der Waals epitaxy. **b** Remote epitaxy.

5. In the third paragraph on page 3, the author discussed α -Te and β -Te without any introduction, so what is the difference between them? The manuscript did not explain in the following contents either.

RESPONSE: Thank you for your patience in reviewing and providing detailed suggestions. We have added a description of the nature and differences in the structure of the Te phase to the Introduction (highlighted in red, page 2, line 28) in revised manuscript.

Tellurium has a unique lattice of chiral chains in which atoms rotate around an axis parallel to the [0001] direction under weak van der Waals forces, where each atom exhibits double coordination, bonding to the other two atoms, and forming a hexagonal lattice. Te possesses a highly anisotropic crystal structure composed of helical chains along the c-axis. These helical chains bind with other chains through van der Waals

interactions. Calculations show that two-dimensional tellurium may have different crystal structures with a variety of bonding configurations. Among them, the α -Te phases exhibit a mixture of triplet and sixfold coordination, while β -Te exhibits a mixture of triplet and quadruple coordination. α -Te configuration corresponds structurally to the 1T configuration commonly adopted for 2D materials known as transition metal dichalcogenides (TMDs), which possess inversion symmetry and semimetallic electronic structures with complex topological states. Therefore, α -Te can be considered a type of transition metal dichalcogenide, with the substitution of the transition metal being a notable alteration. Moreover, α -Te exhibits a layer-dependent bandgap, low exciton binding energy (~ 0.18 eV), extraordinary mobility, high optical absorption, and strong infrared oscillation intensity, which holds broad prospects for applications in sensitive photodetectors (*Appl. Phys. Lett.* 2019, **114**, 092101; *Sci. Bull.* 2018, **63**, 159-168). β -Te is a p-type semiconductor with a bandgap of ~ 0.3 eV, and due to spin-orbit coupling, it possesses a small effective mass and a high hole mobility of up to several thousand $\text{cm}^2 \text{V}^{-1} \text{s}^{-1}$. Meantime, β -Te possesses rich and intriguing characteristics, such as photoconductivity, thermoelectricity, and piezoelectricity, which demonstrates significant potential applications in various fields, including photodetectors, field-effect transistors, piezoelectric devices, modulators, and energy harvesting devices (*Nano-Micro Lett.* 2020, **99**, 1-34).

Furthermore, α -Te arranges along the a-axis by van der Waals forces, while the chains of Te atoms align along the primary c-axis, forming hexagonal nanosheets with thicknesses ranging from tens to hundreds of nanometers. β -Te tends to grow along the c-axis, forming nanoribbons with diameters ranging from tens to hundreds of nanometers. α -Te is the most stable phase (0.05eV/atom more energetically stable than the β -Te) (*2D Mater.* 2019, **6**, 015013).

6. *Lack of basic information related to this research. Such as the structure of the substrate, the growth method of Te, because there are many ways to grow Te, solution based, MBE, and CVD... The CVD growth kinetics is very different from the other methods.*

RESPONSE and CHANGES: Thank you for your insightful opinion. Considering that monolayer WS₂/h-BN (van der Waals substrate) is more favorable for the lateral growth of ultrathin Te crystals due to its high atomic planarity, low surface energy, and fewer defects than SiO₂/Si, we employed a CVD two-step method to investigate the Te phase synthesis process in conjunction with the effect of the gas flow rate on the bonding energy between the atomic clusters and the substrate. Here, we employed WS₂ and h-BN as growth substrates to study the electrical properties of Te in depth. The technical details are as follows:

1. The preparation of monolayer WS₂.

WS₂ monolayers were fabricated on silica/silica substrates by an atmospheric pressure CVD process. First, sodium chloride (NaCl) was ground with an onyx mortar and pestle to a powder particle size of approximately 200 mesh, and then, the ground sodium chloride (NaCl) was homogeneously mixed with tungsten trioxide powder (99.8%, Aladdin) at a rate of 0.05 g : 0.5 g. The mixed powder and a small amount of sulfur powder (99.5%, Alfa Aesar) were then placed in the highest temperature zone and upstream zone of the furnace, respectively. Then, a monolayer of WS₂ was prepared at a flow rate of 200 sccm of Ar (99.995%), a growth temperature of 800 °C, and a growth time of 10 min.

2. The preparation of h-BN.

A few layers of h-BN were prepared on SiO₂/Si₃N₄ by mechanical stripping. The high-quality h-BN crystals were used as the source, mechanically peeled off it with blue tape, repeated approximately 5 times and then pasted them on SiO₂/Si₃N₄ substrate, pressed the blue tape by finger for approximately 2 min and then tore it off, and screened the h-BN nanosheets with the size larger than 10 μm and the thickness less than 20 nm as the growth substrate.

3. CVD growth of ultrathin α -/ β -Te single crystals.

First, the growth substrate, either a single layer of WS₂ obtained by one-step CVD preparation or a few layers of h-BN prepared by mechanical stripping, was placed downstream of a tube furnace with a uniform cooling gradient. A quartz boat containing Te powder (99.99%, Meryer) was then placed in the center of the tube furnace. The α -

Te nanosheets on WS₂/h-BN were obtained by heating the furnace to 430-550 °C (with a cooling zone of 300-450 °C) for 2-20 min at a preset argon flow of 50-125 sccm. The corresponding β -Te nanoribbons on WS₂/h-BN can be obtained by heating the furnace to 430-550 °C (with the cooling zone at 250-450 °C) for 2-20 minutes at an argon flow rate of 125-200 sccm and a growth temperature of 310-510 °C (with the cooling zone at 250-410 °C) for 2-14 minutes. Due to the absence of dangling bonds on the WS₂ and h-BN nanosheet surfaces, resulting in a low charge scattering center density and weaker charge trapping states, Te exhibits superior electrical properties. To further investigate the electrical transport characteristics of Te nanoribbons on WS₂ and h-BN, we employed heavily doped silicon as the global back gate and Au (50 nm) as the contact electrode and patterned the source/drain using EBL to fabricate Te FETs.

Compared to MBE, which typically requires highly matched surfaces, strict temperature and vacuum conditions, vdW epitaxy relies on weak van der Waals forces instead of strict lattice matching or strong chemical bonds, which allows materials to grow on substrates with significant lattice mismatches, and the growth conditions tend to be milder, reducing the complexity and cost of the preparation process. In comparison to solution-based methods, vdW epitaxy allows precise control over the epitaxial growth of 2D materials, which is unaffected by environmental conditions or changes in solution concentrations. This circumvents issues related to impurities or residual solvents, resulting in 2D materials with fewer defects and greater uniformity. Finally, the ability to stack different 2D materials in a controlled manner to form complex heterostructures is a key advantage of vdW epitaxy, which holds significant potential in fields such as electronics and optoelectronics. Additionally, vdW epitaxy is more suitable for large-scale production, providing a more direct pathway for scalability and industrial production. In summary, van der Waals epitaxy has advantages in terms of control, purity, scalability, and precision in creating layered structures, making it a promising technique for producing high-quality films for various applications.

Related discussions have been added to the revised manuscript (highlighted in red, page 21, line 418 and Page22, line 449).

7. Although the authors stated that WS₂ is insulating and does not affect the conduction of Te devices, the authors also mentioned that there is a spatial charge transfer between the two. So a concern arises that whether the performance of Te achieved in this report is the intrinsic performance of Te or it depends on the specific substrates.

RESPONSE: Thank you for your insightful question. We think that the electrical performance of Te in the manuscript is an intrinsic property of Te and does not depend on the specific substrate. The charge transfer between Te and WS₂ is very weak, attributed to the nearly insulating nature of WS₂ synthesized through NaCl-assisted CVD. To demonstrate the inherent electrical performance of Te grown via van der Waals epitaxy in our manuscript we tested β -Te devices prepared on h-BN, as shown in Fig. R11a-b. The current density reached 1527 $\mu\text{A}/\mu\text{m}$ at $V_{\text{ds}} = -4$ V, with a mobility of 567.80 $\text{cm}^2 \text{V}^{-1} \text{s}^{-1}$. Here, we tested β -Te devices prepared on h-BN, as shown in Fig. R11a-b. The current density was as high as 1527 $\mu\text{A}/\mu\text{m}$ at $V_{\text{ds}} = -4$ V, and the mobility was 567.80 $\text{cm}^2 \text{V}^{-1} \text{s}^{-1}$, which once again confirmed that the performance of Te does not rely on a specific substrate.

Furthermore, WS₂ has several advantages, such as an atomically flat surface ($R_{\text{al}} = 0.091$ nm) with a low surface energy and few defects, which are appropriate for the lateral growth of Te and for obtaining thinner Te samples. Under the same growth conditions, the thickness of the β -Te particles was 50-200 nm (Fig. R11d), while the thickness was 0.4-14 nm for WS₂.

Fig. R11 The electrical characteristics of β -Te transistors on h-BN substrates and the morphology of Te grown on WS_2 and SiO_2 substrates. **a, b** Output (**a**) and transfer (**b**) curves (V_{ds} : -0.4 to -4 V) of the β -Te device with a channel length of 1.02 μm on h-BN substrates. **c, d** Optical microscopy images of β -Te nanoribbons on WS_2 (**c**) and SiO_2 (**d**), respectively.

8. Following the above point, one of another biggest questions in the manuscript is whether the Te structures are a pure phase since the Te grows on the WS_2 substrate at a high temperature up to 550 C. Such a high temperature is likely to cause reaction between the substrate and the Te. At least it can be seen from Figure S13b on page 43 that W 4f_{5/2} to W 4f_{7/2} ratio is significantly changed indicating the WS_2 severely changed its chemistry, and it is not a pure phase anymore.

RESPONSE and CHANGES: Thank you for your in-depth questions. As shown in Fig. R12a, we performed XPS tests on Te prepared at 550 °C, and the collected XPS signals were mainly distributed at 583.6 eV and 573.1 eV, corresponding to 3d_{3/2} and 3d_{5/2} of Te, indicating that Te is in a pure phase and does not undergo a chemical reaction with the WS_2 substrate at 550 °C (*npj 2D Mater. Appl.* 2022, **6**, 4; Atomic Number 52 in Handbook of X-ray Photoelectron Spectroscopy). Moreover, we prepared monolayer WS_2 using CVD in one step and then heated the WS_2 again at 550 °C (with the same

parameters as the highest temperature increase Te) to characterize their charge transfer separately. As shown in Fig. R12b, after WS₂ underwent a high-temperature treatment at 550 °C, the ratio between W4f_{5/2} and W4f_{7/2} was 1.15, whereas before the high-temperature treatment, the ratio was 1.18, which indicated that the chemical properties of WS₂ did not undergo significant changes at high temperatures. We speculate that after XPS analysis, the pure WS₂ was exposed to air for an extended period of approximately 10 days, resulting in changes in the chemical properties of the WS₂ itself. This is because WS₂ grown with the assistance of NaCl tends to absorb moisture from the air easily due to residual salt, altering its chemical properties. Therefore, we conducted additional sets of experiments. XPS analysis of WS₂ grown with NaCl-assisted CVD was conducted immediately after 10 days, and the W4f_{5/2}/W4f_{7/2} ratio was 1.18 (0 days) : 2.58 (after 10 days), as shown in Fig. R12c. This result suggested that the variation in the W4f_{5/2}/W4f_{7/2} ratio was attributed to the oxidation of WS₂ itself due to prolonged exposure of NaCl-assisted WS₂ to air rather than to a reaction occurring between WS₂ and Te at high temperatures.

In the supporting information, we have provided corresponding explanations on page 48, line 817 (highlighted in red) in revised manuscript.

Fig. R12. The XPS spectra of Te and monolayer WS₂. **a)** XPS spectrum of the Te 3d core-level region. **b)** Effect of secondary heating of WS₂ on XPS data. **c)** Effect of air exposure of WS₂ on XPS data.

9. Once again, the manuscript lacks introduction to many details. On page 17, it mentioned the h-bn related dielectrics, but how was this realized?

RESPONSE and CHANGES: Thank you for your thoughtful questions. We have added a description of the details of the h-BN in the page 21, line 426; page 22, line

449 of the revised manuscript (highlighted in red).

Here, we report the growth of high-quality Te nanoribbons on atomically flat h-BN for high-performance p-type field-effect transistors (FETs). First, a few layers of h-BN were prepared on SiO₂/Si₃N₄ by mechanical stripping. The high-quality h-BN crystals were used as the source, mechanically peeled off it with blue tape, repeated approximately 5 times and then pasted them on SiO₂/Si₃N₄ substrate, pressed the blue tape by finger for approximately 2 minutes and then tore it off, and screened the h-BN nanosheets with the size larger than 10 μm and the thickness less than 20 nm as the growth substrate. Then, the h-BN on the SiO₂/Si₃N₄ substrate was placed downstream of the tube furnace with a uniform cooling gradient. A quartz boat containing Te powder (99.99%, Meryer) was then placed in the center of the tube furnace. The α-Te nanosheets on h-BN were obtained by heating the furnace to 430-550 °C (with a cooling zone of 300-450 °C) for 2-20 min at a preset argon flow of 50-125 sccm. The corresponding β-Te nanoribbons on h-BN can be obtained by heating the furnace to 430-550 °C (with the cooling zone at 250-450 °C) for 2-20 minutes at an argon flow rate of 125-200 sccm and a growth temperature of 310-510 °C (with the cooling zone at 250-410 °C) for 2-14 minutes. The vdW h-BN dielectric layer possesses an atomically flat surface, providing an ideal platform for the growth of high-quality single-crystal Te nanoribbons. Due to the absence of dangling bonds on the h-BN nanosheet surface, resulting in a low charge scattering center density and weaker charge trapping states, β-Te exhibited superior electrical properties. To further investigate the electrical transport characteristics of Te nanoribbons on h-BN, we employed heavily doped silicon as the global back gate and Au (50 nm) as the contact electrode and patterned the source/drain using EBL to fabricate β-Te FETs, as shown in Fig. R13.

In addition, h-BN stands out as an exceptional dielectric layer in electronic devices due to several key advantages. (1) Its superior insulating properties, with an indirect bandgap of ~6 eV and a dielectric constant typically ranging between 2 and 3, make it an excellent electrical insulator, effectively isolating conductive layers and preventing short circuits within electronic circuits (*Nat. Photonics* 2016, **10**, 262; *Nat. Nanotechnol.* 2010, **5**, 722.). (2) h-BN exhibits impressive chemical stability and good resistance to

water, oxygen, and various corrosive substances, rendering it an ideal choice for applications demanding long-term stability. (3) Its flat, smooth lattice structure makes it suitable for serving as a dielectric layer on flat surfaces, enhancing device performance. (4) The interlayer spacing of h-BN resembles that of other 2D materials, such as graphene, enabling seamless integration and formation of heterostructures. (5) Its remarkable thermal stability ensures structural integrity and performance even under high-temperature conditions, providing an advantage in high-temperature electronic devices. These characteristics collectively position hexagonal boron nitride as a widely favored dielectric material in electronic devices.

Fig. R13 Electrical characterization of Te growing directly on h-BN substrates. **a** Optical images of Te growing directly on h-BN. **b, c** Output (**b**) and transfer (**c**) curves (V_{ds} : -0.4 to -4 V) of the β -Te device with a channel length of 1.02 μm on h-BN substrates.

Reviewer #3

In this work, Zhou et al. realized the phase-engineered synthesis of α - and β -Te single crystals with controlled thicknesses, by delicately modulating the competition between interfacial interaction and cluster interplay. The intricate mechanisms underlying the phase engineering were elucidated through compelling theoretical calculations. The as-synthesized Te, particularly the β -Te, exhibited relatively good Field Effect Transistor (FET) performances. The presented data is of high quality, and the article is well-organized. However, I hesitate to recommend its publication in the journal of Nature Communications due to concerns regarding novelty and practical value. The detailed reasons are listed below:

RESPONSE: We are very grateful to the reviewers for their patience in reviewing our manuscript and for their valuable suggestions and positive comments, such as "The

intricate mechanisms underlying the phase engineering were elucidated through compelling theoretical calculations", "The as-synthesized Te, particularly the β -Te, exhibited relatively good Field Effect Transistor (FET) performances." and "The presented data is of high quality, and the article is well-organized.". We have added experimental data from this work and reorganized the structure of the manuscript to highlight the significance of this work. Below, we respond to each of the comments and point out specific changes and additional information in the manuscript.

1. *The fabrication of β -Te has been frequently reported in recent years through various growth methods on versatile substrates (Nat. Electron. 1, 228-236 (2018), Adv. Mater. 2018, 30, 1803109 (2018), Nano-Micro Lett. 14, 109 (2022), Mater. Today 63, 50-58 (2023)). Some studies have achieved the direct synthesis of β -Te on Si/SiO₂ substrates (Adv. Mater. 2018, 30, 1803109 (2018)) or dielectric materials (e.g., hBN, Nano-Micro Lett. 14, 109 (2022)), making the fabrication process more industrially compatible. Moreover, certain reports on the FET performances of β -Te, including mobility and on-off ratio, surpass those presented in this work. The growth methods, substrates, and FET performances in this study do not showcase significant advantages over these prior reports.*

RESPONSE: Thank you for your insightful questions. The article's innovation lies in the following aspects:

(1) If we use the formula for 2D materials reported in the literature to calculate the mobility of β -Te nanoribbons (Nat. Electron. 2018, **1**, 228-236; Adv Mater.2018; **30**, 1803109; Mater. Today 2023, **63**, 50-58; and Nano-Micro Lett. 2022, **14**, 109). The maximum value reaches $3752.27 \text{ cm}^2 \text{ V}^{-1} \text{ s}^{-1}$ (Table R3), with a current density of $\sim 1.50 \text{ mA } \mu\text{m}^{-1}$ at $V_{\text{ds}} = -1.5 \text{ V}$, which is significantly greater than the reported mobility of β -Te ($1370 \text{ cm}^2 \text{ V}^{-1} \text{ s}^{-1}$ with a current density of $\sim 10 \text{ } \mu\text{A } \mu\text{m}^{-1}$ at $V_{\text{ds}} = 0.5 \text{ V}$ in Nano-Micro Lett. 2022, **14**, 109; $1755 \text{ cm}^2 \text{ V}^{-1} \text{ s}^{-1}$ with a current density of $\sim 47.2 \text{ } \mu\text{A } \mu\text{m}^{-1}$ at $V_{\text{ds}} = 100 \text{ mV}$ in Mater. Today 2023, **63**, 50-58). As our β -Te width is smaller than the dielectric layer thickness, to ensure the accuracy of the electrical performance and avoid overstating device capabilities, we utilized nanowire formulas in this manuscript to

calculate a mobility of $\sim 690.7 \text{ cm}^2 \text{ V}^{-1} \text{ s}^{-1}$, which is still comparable to the reported mobility ($\sim 700 \text{ cm}^2 \text{ V}^{-1} \text{ s}^{-1}$ in *Nat. Electron.* 2018, **1**, 228-236, *Adv Mater.* 2018, **30**, 1803109). Additionally, according to the International Roadmap for Devices and Systems (IRDS), for semiconductor metal-oxide-semiconductor (MOS) field-effect transistors (FETs), the final power supply voltage will be $\leq 1.0 \text{ V}$. At $V_{\text{ds}} = -1 \text{ V}$, the current density of the β -Te FET in our article is $1.27 \text{ mA } \mu\text{m}^{-1}$, which is significantly greater than the reported β -Te current density ($\sim 0.85 \text{ mA } \mu\text{m}^{-1}$ at $V_{\text{ds}} = -1 \text{ V}$).

(2) Furthermore, because the synthesized β -Te particles are relatively thick, their ON/OFF ratio is very small ($\sim 10^1$ - 10^2 in *Adv. Mater.* 2018, **30**, 1803109; *Mater. Today* 2023, **63**, 50-58). Although Wu et al. reported a current density of $1.06 \text{ mA } \mu\text{m}^{-1}$ for 11.1 nm thick β -Te at $V_{\text{ds}} = -1.4 \text{ V}$, their ON/OFF ratio was only 10^1 (*Nat. Electron.* 2018, **1**, 228-236). In contrast, the majority of our β -Te devices maintain on-off ratios of 10^3 - 10^4 . Particularly in narrow-channel devices (46 nm), at $V_{\text{ds}} = -1.5 \text{ V}$, the current density reaches $\sim 1.5 \text{ mA } \mu\text{m}^{-1}$, while the on-off ratio still remains at 10^3 .

(3) Most of the existing 2D materials suffer from either low carrier mobility (such as MoS_2 and MoTe_2) or poor air stability (such as BP). Therefore, identifying a 2D material that combines the advantages of high carrier mobility, good air stability, and easy fabrication is vital to the development of electronic devices in the next generation. Our β -Te nanoribbons exhibit excellent air-stable p-type electrical performance, while our α -Te nanosheets demonstrate outstanding air-stable metallic properties, which enables the possibility of achieving metal-semiconductor heteroepitaxial integration within the same atomic plane by leveraging phase engineering of Te, thereby achieving atomic-scale metal-semiconductor contacts and reducing contact resistance. As a result, the realization of Te phase engineering lays the foundation for enhancing optoelectronic devices and facilitating large-scale, high-performance back-end-of-line integrated circuits.

(4) We can grow large-area, highly oriented Te single crystals. The highly single-orientation (90%) epitaxy of β -Te nanoribbons on monolayer 2-inch MoS_2 provides a reliable path for the future preparation of large-area, high-quality, and scalable 2D Te arrays and single-crystal films. This has laid a solid foundation for the future integration

and industrial application of 2D Te devices.

Table R3 Mobility of β -Te devices under different computational formulas.

Number	L (μm)	W (μm)	$ V_{\text{ds}} $	$dI_{\text{ds}}/dV_{\text{g}}$	Updated $\mu(\text{cm}^2 \text{V}^{-1} \text{s}^{-1})$	μ of Eq. 4 ($\text{cm}^2 \text{V}^{-1} \text{s}^{-1}$)
1	5.181	0.07621	1	2.03×10^{-7}	1198.27	218.57
2	5.522	0.1414	1	2.78×10^{-7}	942.35	278.98
3	5.522	0.1414	1.5	4.04×10^{-7}	914.16	211.87
4	4.839	0.058	0.5	6.15×10^{-8}	886.25	149.44
5	2.961	0.0468	2	3.93×10^{-7}	1078.55	136.85
6	3.949	0.086	1	1.75×10^{-7}	698.67	135.23
7	2.88	0.0192	0.1	1.75×10^{-8}	2286.52	149.26
8	3.75	0.0513	2	5.2×10^{-7}	1652.05	223.45
10	3.064	0.0189	0.4	6.37×10^{-8}	2235.97	144.92
11	3.979	0.0886	0.5	1.2×10^{-7}	936.82	185.09
12	3.198	0.06168	0.5	1.14×10^{-7}	1023.43	158.27
13	2.704	0.02155	0.4	1.01×10^{-7}	2760.46	196.73
14	4.729	0.0829	0.7	1.55×10^{-7}	1098.37	207.38
15	6.152	0.1316	0.4	1.27×10^{-7}	1287.59	331.32
16	4.853	0.1108	0.1	1.96×10^{-8}	745.35	171.21
17	6.09	0.0864	0.1	1.3×10^{-8}	794.96	154.72
18	6.224	0.0978	0.7	2.23×10^{-7}	1758.99	176.17
19	5.968	0.2128	0.1	3.17×10^{-8}	773.06	269.03
20	6.228	0.111	1.5	6.32×10^{-7}	2055.67	350.54
21	5.898	0.1972	2.5	2.0×10^{-6}	2080.60	690.69
22	5.969	0.0956	0.5	2.12×10^{-7}	2306.37	254.27
23	5.889	0.09618	3	1.65×10^{-6}	2924.79	621.89
24	5.972	0.1082	3	7.01×10^{-7}	1121.47	185.92
25	5.397	0.0877	2.5	7.75×10^{-7}	1658.88	211.19

26	4.361	0.0690	1	1.29×10^{-7}	707.25	117.98
27	3.009	0.149	2	4.61×10^{-7}	405.12	112.57
28	0.9182	0.01629	0.1	4.78×10^{-8}	2342.86	135.02
29	0.9278	0.01288	1	4.09×10^{-7}	2561.91	122.99
30	0.9	0.047	0.4	4.33×10^{-7}	1802.5	228.94
31	0.92	0.027	0.4	2.52×10^{-7}	1866.67	157.98
32	0.9931	0.04962	0.7	5.40×10^{-7}	1342.56	177.56
33	0.975	0.03031	1	6.32×10^{-7}	1767.82	163.07
34	0.9193	0.02477	1.5	9.57×10^{-7}	2058.99	163.34
35	1.022	0.036	1	1.52×10^{-6}	3752.27	567.80

2. Although the authors successfully realize the phase engineering of α - and β -Te using WS₂ as a growth substrate, the potential applications or intriguing properties of α -Te seem limited. The synthesis of α -Te itself has also been previously reported. The significance of this phase engineering strategy warrants further consideration.

RESPONSE and CHANGES: Thank you for your suggestion. In the literature, a polycrystalline phase of Te that coexists with the α -, β -, and γ -phases was previously reported, but a pure phase of α -Te has not been synthesized (*2D Mater.* 2019, **6**, 015013).

(1) The potential applications and intriguing properties of α -Te

α -Te features a thickness-dependent bandgap, extraordinary mobility (three times that of MoS₂), strong light absorption, high stretchability, and excellent environmental stability (*Appl. Phys. Lett.* 2019, **114**, 092101). In addition, α -Te has a relatively low exciton binding energy (~0.18 eV), high optical absorption, and oscillator strength in the infrared region. The absorbance exhibits layer-dependent behavior, with absorption efficiency increasing as the α -Te thickness decreases. Its high mobility and optical absorption suggest potential applications in sensitive photodetectors (*Sci. Bull.* 2018, **63**, 159-168). Furthermore, the nearly isotropic, in-plane optical absorption of α -Te results in stronger absorption compared to β -Te (*Nano-Micro Lett.* 2020, **99**, 1-34).

(2) The significance of phase engineering

Two-dimensional (2D) Te may have different crystal structures with a variety of bonding configurations. Among them, the α -Te phases exhibit a mixture of triplet and sixfold coordination, while β -Te exhibits a mixture of triplet and quadruple coordination, which have very different physical and electronic properties. The α -Te configuration corresponds structurally to the 1T configuration commonly adopted for 2D materials known as transition metal dichalcogenides (TMDs), which possess inversion symmetry and semimetallic electronic structures with complex topological states. Moreover, α -Te exhibits a layer-dependent bandgap, low exciton binding energy (~ 0.18 eV), extraordinary mobility, high optical absorption, and strong infrared oscillation intensity, which holds broad prospects for applications in sensitive photodetectors (*Appl. Phys. Lett.* 2019, **114**, 092101; *Sci. Bull.* 2018, **63**, 159-168). β -Te is a p-type semiconductor with a bandgap of ~ 0.3 eV, and due to spin-orbit coupling, it possesses a small effective mass and a high hole mobility of up to several thousand $\text{cm}^2 \text{V}^{-1} \text{s}^{-1}$. Meantime, β -Te possesses rich and intriguing characteristics, such as photoconductivity, thermoelectricity, and piezoelectricity, which demonstrates significant potential applications in various fields, including photodetectors, field-effect transistors, piezoelectric devices, modulators, and energy harvesting devices (*Nano-Micro Lett.* 2020, **99**, 1-34). α -Te is the most stable phase (0.05 eV/atom more energetically stable than the β -Te) (*2D Mater.* 2019, **6** 015013). The dynamic control of α -Te and β -Te phase engineering can reveal the competition, coexistence, and cooperation among different crystal structures as well as the interactions between different physical properties. Therefore, phase engineering between semimetallic α -Te and semiconducting β -Te polymorphs holds significant importance for widespread device applications, such as phase-engineering memory devices, high-performance transistors, reconfigurable circuits, and topological transistors, under atomically thin limits. To date, p-type β -Te materials with excellent electrical properties have been successively synthesized using various methods, such as hydrothermal synthesis, molecular beam epitaxy (MBE), thermal evaporation, atomic layer deposition (ALD), and chemical vapor deposition (CVD). Although Wu et al. synthesized β -Te nanosheets

with mobilities of approximately $700 \text{ cm}^2 \text{ V}^{-1} \text{ s}^{-1}$ and a significant ON-state current density of $1.06 \text{ mA } \mu\text{m}^{-1}$ at $V_{\text{ds}} = -1.4 \text{ V}$ using a substrate-free solution, their ON/OFF ratio was only 10^1 . In addition, Zhao et al. reported the fabrication of β -Te thin films by thermal evaporation at room temperature with an average mobility of $21.1 \text{ cm}^2 \text{ V}^{-1} \text{ s}^{-1}$, an ON-state current of approximately $0.001 \text{ mA } \mu\text{m}^{-1}$ and an ON/OFF current ratio of $\sim 10^4$. However, these previous reports have been limited to the synthesis of β -Te, and phase engineering between semimetallic α -Te and semiconducting β -Te has not been reported to date, especially at the monolayer limit. Exploring phase transition competition at the atomic scale is crucial because it involves significantly enhanced electron-phonon and electron-electron interactions. Here, we designed an atomic cluster density- and interface-guided multiple controllable strategy for the phase- and thickness-controlled synthesis of monolayer α -Te nanosheets and β -Te nanoribbons. The α -Te nanosheets exhibit a transition from a metal to an n-type semiconductor, whereas the β -Te nanoribbons are always p-type semiconductors as the thickness decreases, and both of them have good air stability after several months. Furthermore, the reduced thickness restricts the transport of carriers in the atomic channel and causes the β -Te nanoribbon to exhibit an exceptional ON-state current density of $\sim 1527 \text{ } \mu\text{A } \mu\text{m}^{-1}$ and a mobility as high as $\sim 690.7 \text{ cm}^2 \text{ V}^{-1} \text{ s}^{-1}$ at room temperature, which suggests that 2D Te could be applied in electronic devices that offer higher performance than those that use MoS_2 .

By employing the continuous vdW epitaxy growth of different phases of Te, achieving heteroepitaxial integration of metal-semiconductor states within the same atomic plane has become feasible, enabling atomic-scale metal-semiconductor contacts and reducing contact resistance. Therefore, the realization of Te phase engineering has paved the way for enhancing the performance of optoelectronic devices, wearable electronics, synaptic devices, and friction-based nanogenerators, laying the groundwork for large-scale, high-performance integrated circuits.

In addition, we have added a description of the significance of α -Te, β -Te, and phase engineering in the revised manuscript (highlighted in red, page 2-3 and page 20, line 407).

3. In the last sentence of page 3, the authors stated that synthesis of single-layer β -Te had not yet been achieved. However, based on my knowledge, this has been realized by Huang et al using an MBE route (*Nano Lett.* 17, 4619–4623 (2017)) and Wang et al. via a solution process (*Nat. Electron.* 1, 228-236 (2018)). It is advisable for the authors to scrutinize the accuracy of this statement.

RESPONSE and CHANGES: We thank the reviewers for their patience in reviewing our work, and we have deleted and changed the description of monolayer β -Te synthesis in the latest sentence of page 3, line 53 (highlighted in red) in the revised version.

(1) Although Guo et al. obtained monolayer Te on the surface of graphene on a 6H-SiC (0001) substrate by MBE (*Nano Lett.* 2017, **17**, 4619-4623), its excellent electrical properties have not been studied. Moreover, compared to MBE, which typically requires highly matched surfaces, strict temperature and vacuum conditions, CVD epitaxy relies on weak van der Waals forces instead of strict lattice matching or strong chemical bonds, which allows materials to grow on substrates with significant lattice mismatches, and the growth conditions tend to be milder, reducing the complexity and cost of the preparation process.

(2) If we use the formula reported in the literature to calculate the mobility of β -Te nanoribbons (*Nat. Electron.* 2018, **1**, 228-236); the maximum value reaches $3752.27 \text{ cm}^2 \text{ V}^{-1} \text{ s}^{-1}$, as shown in Table R3, which is significantly greater than the reported mobility of β -Te ($\sim 700 \text{ cm}^2 \text{ V}^{-1} \text{ s}^{-1}$ in *Nat. Electron.* 2018, **1**, 228-236). Additionally, according to the International Roadmap for Devices and Systems (IRDS), for semiconductor metal-oxide-semiconductor (MOS) field-effect transistors (FETs), the final power supply voltage will be $\leq 1.0 \text{ V}$. At $V_{ds} = -1 \text{ V}$, the current density of the β -Te FET in our article is $1.27 \text{ mA } \mu\text{m}^{-1}$, which is significantly higher than the reported β -Te current density ($\sim 0.85 \text{ mA } \mu\text{m}^{-1}$ at $V_{ds} = -1 \text{ V}$). Although Wu et al. reported a current density of $1.06 \text{ mA } \mu\text{m}^{-1}$ for 11.1 nm thick β -Te at $V_{ds} = -1.4 \text{ V}$, their ON/OFF ratio was only 10^1 (*Nat. Electron.* 2018, **1**, 228-236). In contrast, the majority of our β -Te devices maintain on-off ratios of 10^3 - 10^4 . Particularly in narrow-channel devices (46 nm), at $V_{ds} = -1.5 \text{ V}$, the current density reaches $\sim 1.5 \text{ mA } \mu\text{m}^{-1}$, while the ON/OFF ratio

still remains at 10^3 . Furthermore, in comparison to solution-based methods, CVD epitaxy allows precise control over the epitaxial growth of 2D materials, which is unaffected by environmental conditions or changes in solution concentrations. This circumvents issues related to impurities or residual solvents, resulting in 2D materials with fewer defects and greater uniformity. Additionally, CVD epitaxy is more suitable for large-scale production, providing a more direct pathway for scalability and industrial production. In summary, van der Waals epitaxy has advantages in terms of control, purity, scalability, and precision in creating layered structures, making it a promising technique for producing high-quality films for various applications.

4. In Figure 1c, the authors compared the formation energies of α - and β -Te by calculating the formation energy of Te atoms combined with WS₂ substrate to form α -Te nanosheets (α -Te@WS₂) and the formation energy of Te atoms combined with each other to form β -Te clusters with respect to atomic cluster. Regarding β -Te, the influence of interfacial interaction between Te atoms and the WS₂ substrate on the growth process should also be considered. However, it appears that the authors did not account for this factor in their calculations.

RESPONSE and CHANGES: We thank the reviewers for their patience in reviewing and providing valuable suggestions. We performed additional calculations to study the influence of interfacial interactions between Te atoms and the WS₂ substrate on the growth process. The formation energy E_f of the two situations, including when the Te atom may bind directly to the WS₂ substrate or extend to the Te cluster, was calculated by using formula (2) in the manuscript. As shown in Fig. R14, the formation energy of the Te atom directly binding with the WS₂ substrate is -254.7 kJ/mol, while the formation energies of the Te atom adsorbed in the armchair and zigzag directions of the cluster are -256.0 and -229.2 kJ/mol, respectively. This further suggests that β -Te is more inclined to form nanoribbons along the armchair direction, and agrees well with our experimental results. The consideration here is the growth of the individual Te atom on the substrate and cluster, while in Fig. 1e, multiple Te atoms are considered. Therefore, the formation energy obtained here is higher than that in Fig. 1e, further

indicating a stronger interaction between Te atoms.

We have added the influence of interfacial interaction between Te atoms and the WS₂ substrate on the growth process to Supplementary Fig. 2 and an additional description in the revised manuscript (highlighted in red, page 8, line 151 and page 34, line 669).

Fig. R14 Formation energies of Te atoms binding with the WS₂ substrate and the β -Te cluster at high flow rates. The insets are side views and top views of the atomic structure, and the red rectangular box represents the β -Te clusters.

5. The authors solely conducted AFM studies to substantiate the stability of α -Te. Additional characterizations may be warranted, particularly given the typical poor air stability associated with metallic 2D materials.

RESPONSE and CHANGES: We thank the reviewers for their valuable comments. Based on your suggestions, we investigated the stability of the electrical properties of α -Te devices with a metallic thickness of approximately 2.9 nm. Fig. R15 shows the output and transfer characteristics of the α -Te device, which were measured immediately (Fig. R15a-b) and after 6 months (Fig. R15c-d). Table R4 shows the α -Te device current values at $V_{ds} = 1$ V for different gate voltages. The metal α -Te devices still maintain almost the same electrical properties over a period of 6 months, which demonstrates the good stability of the α -Te devices.

We have added these additional output characteristics and transfer characteristics of the α -Te devices, measured after 6 months, to Fig. 4h and Supplementary Fig. 17, and an additional description in the revised manuscript (highlighted in red, page 17, line 340).

Fig. R15 The stability of the α -Te device. **a, c** Output curves at 0 (a) and 6 months (c), respectively. **b, d** Transfer curves at 0 (b) and 6 months (d), respectively.

Table R4 Current values of α -Te devices at different gates when $V_{ds} = 1$ V.

	-40 V	-20 V	0 V	20 V	40 V
0 months	4.31×10^{-4} A	4.39×10^{-4} A	4.52×10^{-4} A	4.67×10^{-4} A	4.80×10^{-4} A
After 6 months	4.00×10^{-4} A	4.08×10^{-4} A	4.18×10^{-4} A	4.29×10^{-4} A	4.39×10^{-4} A

6. The last sentence on page 7 was repeated twice.

RESPONSE and CHANGES: Thank you for your careful reading and note. In the revised version, we have removed duplicate sentences and streamlined the description on page 8 (highlighted in red, page 8, line 163).

REVIEWERS' COMMENTS

Reviewer #1 (Remarks to the Author):

I have read this manuscript for a second time, and while the authors have made progress in improving the quality of data and technical details, I still withstand my previous conclusion and refrain from recommending for publication. The major flaw of the paper is lack of novelty, since the outcome of the paper neither provides much new, insightful understanding of tellurium's properties significantly beyond previous work, nor yields any meaningful nanostructure that can propel the future research or commercial applications for this material. I give authors credit for the high mobility of the material and systematic investigation of growth conditions, and I think it is more appropriate to transfer the manuscript to a more specialized journal.

Here below are my comments to the author's revision.

(1) The question is addressed.

(2) The question is addressed.

(3) I believe the community has been using different nomenclature for different Te phases. In bulk Te, each Te atom only has 2 covalent bonds (Nano Lett. 2017, 17, 6, 3965), forming into a 3D trigonal lattice. However in Fig. 1a the beta-Te obviously has 3-4 bonds for each Te site, and the unit cell forms a 2D rectangular lattice (Phys. Rev. Lett. 2017, 119, 106101; Nature 2017, 552, 40-41; Phys. Rev. B 2019, 99, 195436; Appl. Phys. Lett. 2019, 115, 151104). As far as I know, this 2D beta-Te has not been reported experimentally. Obviously, beta-Te and bulk Te should be two distinct phases. Beta-Te should not be the monolayer version of bulk Te. Can the authors clarify which group does your "beta-Te" belong to?

(4) The question is addressed.

(5) The mobility (eq. 1) is calculated for semiconductor devices in the linear region, and should not be applied here. In a semiconductor device, when operated in linear region, the current is determined by drift velocity, which can be associated with the carrier mobility. This is how field-effect mobility is derived in the first place. However in this case the material is more like a metal, or semimetal, rather than a semiconductor. This equation 1 no longer holds. In other words, the transconductance does not reflect the carrier mobility, ie how fast carriers move through the channel; the field effect is mainly because of the change of carrier density. The authors can report transconductance for alpha-Te but not mobility by simply applying this equation.

(6) The question is addressed.

(7) The question is addressed.

(8) The question is addressed.

(9) The authors have addressed my original question. However, looking through Table R3, the authors tend to use a high V_{ds} to calculate mobility. Is there a reason to do so? From Fig. R5, most devices are already in saturation regime when $V_{ds} > 1V$. The authors should use a low V_{ds} when possible to extract mobility to ensure the device operates in the linear regime.

(10) The question is addressed.

(11) The question is addressed.

Reviewer #2 (Remarks to the Author):

The authors have answered all my questions and I think this paper is good to go.

Reviewer #3 (Remarks to the Author):

The authors have addressed my concerns, and I think this article now meets the criteria for publication in Nature Communications.

Reviewers' comments and our responses

Reviewer #1:

I have read this manuscript for a second time, and while the authors have made progress in improving the quality of data and technical details, I still withstand my previous conclusion and refrain from recommending for publication. The major flaw of the paper is lack of novelty, since the outcome of the paper neither provides much new, insightful understanding of tellurium's properties significantly beyond previous work, nor yields any meaningful nanostructure that can propel the future research or commercial applications for this material. I give authors credit for the high mobility of the material and systematic investigation of growth conditions, and I think it is more appropriate to transfer the manuscript to a more specialized journal.

RESPONSE: We thank the reviewers for their previous patient analysis and valuable suggestions to improve the quality of our manuscript and for recognizing our adjustments and additions to the content of the manuscript with the positive words "*the authors have made progress in improving the quality of data and technical details*". In addition, the reviewers gave us a high rating of "*I give authors credit for the high mobility of the material and systematic investigation of growth conditions*" for our research work. With the help of the reviewer, we have significantly improved the content and meaning of our report titled "Phase-Engineered Synthesis of Atomically Thin Te Single Crystals with High ON-State Currents".

The article's innovation lies in the following aspects:

(1) Despite the gradual improvement in theoretical and experimental studies on tellurium, the current research reports still mainly focus on the photoelectric properties of β -Te only, and there are few studies on phase control. Therefore, our work advances the development of tellurium in the field of 2D electronic integration through phase engineering control.

(2) If we use the formula reported in the literature to calculate the mobility of β -Te nanoribbons (*Nat. Electron.* 2018, **1**, 228-236; *Adv. Mater.* 2018, **30**, 1803109; *Mater.*

Today 2023, **63**, 50-58). The maximum value reaches $3752.27 \text{ cm}^2 \text{ V}^{-1} \text{ s}^{-1}$; this value is significantly greater than the reported mobility of β -Te ($1755 \text{ cm}^2 \text{ V}^{-1} \text{ s}^{-1}$ in *Mater. Today* 2023, **63**, 50-58). As our β -Te width is smaller than the dielectric layer thickness, to ensure the accuracy of the electrical performance and avoid overstating device capabilities, we utilized the more conservative calculation formula in this manuscript to calculate a mobility of $\sim 690.7 \text{ cm}^2 \text{ V}^{-1} \text{ s}^{-1}$, which is still comparable to the reported mobility ($\sim 700 \text{ cm}^2 \text{ V}^{-1} \text{ s}^{-1}$ in *Nat. Electron.* 2018, **1**, 228-236; *Adv. Mater.* 2018, **30**, 1803109). Additionally, according to the International Roadmap for Devices and Systems (IRDS), for semiconductor metal-oxide-semiconductor (MOS) field-effect transistors (FETs), the final power supply voltage will be $\leq 1.0 \text{ V}$. At $V_{\text{ds}} = -1 \text{ V}$, the current density of the β -Te FET in our article is $1.27 \text{ mA } \mu\text{m}^{-1}$, which is significantly higher than the reported β -Te current density ($\sim 0.85 \text{ mA } \mu\text{m}^{-1}$ at $V_{\text{ds}} = -1 \text{ V}$).

(3) Furthermore, because the synthesized β -Te particles are relatively thick, their ON/OFF ratio is very small ($\sim 10^1$ - 10^2 in *Adv. Mater.* 2018, **30**, 1803109; *Mater. Today* 2023, **63**, 50-58). Although Wu et al. reported a current density of $1.06 \text{ mA } \mu\text{m}^{-1}$ for 11.1 nm thick β -Te at $V_{\text{ds}} = -1.4 \text{ V}$, their ON/OFF ratio was only 10^1 (*Nat. Electron.* 2018, **1**, 228-236). In contrast, the majority of our β -Te devices maintain ON/OFF ratios of 10^3 - 10^4 . Particularly in narrow-channel devices (46 nm), at $V_{\text{ds}} = -1.5 \text{ V}$, the current density reaches $\sim 1.5 \text{ mA } \mu\text{m}^{-1}$, while the ON/OFF ratio still remains at 10^3 .

(4) We have demonstrated that the synthesis of our material may not be limited by the choice of substrate. The highly single-orientation (90%) epitaxy of β -Te nanoribbons on monolayer 2-inch MoS_2 provides a reliable path for the future preparation of large-area, high-quality, and scalable 2D Te arrays and single-crystal films. This has laid a solid foundation for the future integration and industrial application of 2D Te devices.

(5) Most of the existing 2D materials suffer from either low carrier mobility (such as MoS_2 and MoTe_2) or poor air stability (such as BP). Therefore, identifying a 2D material that combines the advantages of high carrier mobility, good air stability, and easy fabrication is vital to the development of electronic devices in the next generation. Our β -Te nanoribbons exhibit excellent air-stable p-type electrical performance, while

α -Te nanosheets demonstrate outstanding air-stable semimetal properties, which enables the possibility of achieving metal-semiconductor heteroepitaxial integration within the same atomic plane by leveraging phase engineering of Te, thereby achieving atomic-scale metal-semiconductor contacts and reducing contact resistance. As a result, the realization of Te phase engineering lays the foundation for enhancing optoelectronic devices and facilitating large-scale, high-performance back-end-of-line integrated circuits.

(3) I believe the community has been using different nomenclature for different Te phases. In bulk Te, each Te atom only has 2 covalent bonds (*Nano Lett.* 2017, 17, 6, 3965), forming into a 3D trigonal lattice. However in Fig. 1a the beta-Te obviously has 3-4 bonds for each Te site, and the unit cell forms a 2D rectangular lattice (*Phys. Rev. Lett.* 2017, 119, 106101; *Nature* 2017, 552, 40-41; *Phys. Rev. B* 2019, 99, 195436; *Appl. Phys. Lett.* 2019, 115, 151104). As far as I know, this 2D beta-Te has not been reported experimentally. Obviously, beta-Te and bulk Te should be two distinct phases. Beta-Te should not be the monolayer version of bulk Te. Can the authors clarify which group does your "beta-Te" belong to?

RESPONSE: We thank the reviewer for the careful review of our manuscript and valuable discussion. The monolayer β -Te belongs to the $P2/m$ symmetry group (*Phys. Rev. B* 2023, **107**, 155420). However, monolayer β -Te and multilayer β -Te do not belong to the same space group, and our XRD characterization pertains to multilayer 2D β -Te. To avoid unnecessary misunderstandings, we have provided clarifications in the manuscript (page 9, Line190).

As shown in Figure R1, when we removed the "additional interchain bonding" mentioned by the reviewer, the 1D van der Waals structure with threefold screw symmetry became visually clearer (Figure R1c). In two-dimensional situations, the structure may undergo slight deformation, with some bond lengths experiencing minor changes compared to those in the bulk. Nevertheless, the 1D van der Waals structure in the monolayer β -Te here is almost the same as that in the bulk. However, to align with the previous literature (*Phys. Rev. Lett.* 2017, **119**, 106101; *Nature* 2017, **552**, 40-41;

Phys. Rev. B 2019, **99**, 195436; *Appl. Phys. Lett.* 2019, **115**, 151104), we chose to display these bonds in Figure 1a of the manuscript.

Fig. R1 Prospective view of the atomic structure of monolayer β -Te. **a** Top view, **b** Front view, **c** Left-side view. The black cubic box indicates the unit cell, and this figure is a three-dimensional perspective, making the cubic box seem distorted. The purple dashed lines represent the removed "additional interchain bonding".

(5) The mobility (eq. 1) is calculated for semiconductor devices in the linear region and should not be applied here. In a semiconductor device, when operated in linear region, the current is determined by drift velocity, which can be associated with the carrier mobility. This is how field-effect mobility is derived in the first place. However in this case the material is more like a metal, or semimetal, rather than a semiconductor. This equation 1 no longer holds. In other words, the transconductance does not reflect the carrier mobility, ie how fast carriers move through the channel; the field effect is mainly because of the change of carrier density. The authors can report transconductance for alpha-Te but not mobility by simply applying this equation.

RESPONSE: We thank the reviewers. We obtained a maximum mobility of $74.7 \text{ cm}^2\text{V}^{-1}\text{s}^{-1}$ at -60 V in our manuscript, which is consistent with the calculation methods for the mobility of previously reported semimetallic two-dimensional materials (*ACS Nano* 2018, **12**, 4055-4061; *Nat. Nanotech.* 2010, **4**, 487-496). Considering the reviewer's suggestions and to make the manuscript more scientifically accurate, we have modified the description of the mobility of α -Te in the latest revised manuscript

to a description of the transconductance of α -Te.

We have changed the corresponding information in the latest revision of the manuscript (highlighted in red, page 13, line 286).

(9) The authors have addressed my original question. However, looking through Table R3, the authors tend to use a high V_{ds} to calculate mobility. Is there a reason to do so? From Fig. R5, most devices are already in saturation regime when $V_{ds} > 1V$. The authors should use a low V_{ds} when possible to extract mobility to ensure the device operates in the linear regime.

RESPONSE: We thank the reviewers for agreeing with our earlier questions and for patiently guiding us to think more deeply about the mobility calculation. We did not use a high V_{ds} to calculate mobility. In fact, most of the mobility calculations we performed were obtained at small V_{ds} , as shown in Table R1, which illustrates the mobility and the corresponding V_{ds} values for the numbers R1-R8 in the corresponding Fig. R2. In long-channel devices (3-6.3 μm), when $V_{ds} \geq 4.99$ V, the device still remains unsaturated and operates in the linear region (Fig. R3). In addition, following the reviewer's suggestions, we carefully examined the mobility of all the β -Te devices, ensuring that all the calculations were performed in the linear region with relatively small V_{ds} .

Fig. R2 Electrical characterization of β -Te transistors. a-I Output (a, c, e, g, i, k) and transfer (b, d, f, h, j, l) curves of β -Te transistors of different thicknesses on WS_2/SiO_2 substrates. m-p Output (m, o) and transfer (n, p) curves of β -Te nanoribbon FETs on h-BN/80 nm thick $\text{Si}_3\text{N}_4/\text{Si}$ substrates with thicknesses of 5.2 nm and 14.3 nm, respectively. q-t Output (q, s) and transfer (r, t) curves of the β -Te devices on h-BN/80-nm $\text{Si}_3\text{N}_4/\text{Si}$ substrates with channel lengths of ~ 66 nm and ~ 46 nm, respectively. In the Output curves of β -Te transistors: in B1-B6 and B9-B10, the various gate voltages from 60 V to -60 V (-10 V step); and in B7-B8, the various gate voltages from 40 V to -40 V (-5 V step). In the transfer curves of β -Te transistors: in B1- B10, the various gate voltages from -0.1 V to -6 V correspond to different colored lines: red, -0.1 V; blue, -0.4 V; dark blue, -0.7 V; purple, -1 V; pink, -1.5 V; azure blue, -2 V; gray, -2.5 V; light purple, -3 V; orange, -3.5 V; blue-gray, -4 V; light pink, -5 V; olive-green, -6 V.

Fig. R3 I_{ds} - V_{ds} of β -Te transistors with $L = 3 \mu\text{m}$.

Table R1 Detailed electrical data for β -Te devices on SiO_2 and Si_3N_4 corresponding to the numbers B1-B8 in Fig. R2.

Sample	L (μm)	Mobility	
		μ ($\text{cm}^2 \text{V}^{-1} \text{s}^{-1}$)	V_{ds} (V)
B1	0.92	135.02	-0.1
B2	0.92	157.98	-0.4
B3	0.98	132.88	-0.4
B4	0.78	242.68	-0.1
B5	0.90	228.94	-0.4
B6	0.99	159.47	-0.1
B7	1.22	387.82	-0.4
B8	1.02	632.25	-0.4